# A universal packaging substrate for mechanically stable assembly of stretchable electronics

Yan Shao[1,2,5], Jianfeng Yan[1,5], Yinglin Zhi[1], Chun Li[1], Qingxian Li[3], Kaimin Wang[1], Rui Xia[1], Xinyue Xiang[1], Liqian Liu[1], Guoli Chen[1], Hanxue Zhang[1], Daohang Cai[1], Haochuan Wang[1], Xing Cheng [1], Canhui Yang [3], Fuzeng Ren [1] ✉ & Yanhao Yu [1,4] ✉

Stretchable electronics commonly assemble multiple material modules with varied bulk moduli and surface chemistry on one packaging substrate. Preventing the strain-induced delamination between the module and the substrate has been a critical challenge. Here we develop a packaging substrate that delivers mechanically stable module/substrate interfaces for a broad range of stiff and stretchable modules with varied surface chemistries. The key design of the substrate was to introduce module-specific stretchability and universal adhesiveness by regionally tuning the bulk molecular mobility and surface molecular polarity of a near-hermetic elastic polymer matrix. The packaging substrate can customize the deformation of different modules while avoiding delamination upon stretching up to 600%. Based on this substrate, we fabricated a fully stretchable bioelectronic device that can serve as a respiration sensor or an electric generator with an in vivo lifetime of 10 weeks. This substrate could be a versatile platform for device assembly.

The operation of stretchable electronics relies on the integration of different material modules[1–3]. Directly fabricating these modules on one packaging substrate is a straightforward option, but is an intricate task for manufacturing[4–6]. A feasible path would be separately fabricating the modules and assembling them together through packaging, akin to conventional microelectronics[7,8]. This way brings a new challenge: modules are prone to delaminate from the packaging substrate under stretching, especially for modules undergoing large deformation during their operation (e.g., strain sensors and transistors)[9,10]. To improve the mechanical stability of assembly, the stretchable packaging substrate should universally adhere to and mechanically comply with different modules that were made from metals, ceramics, plastics, and elastomers.

The adhesions between the module and the substrate can be formed through chemical bonding or topological entanglement for chemically alike interfaces, broadly covering silicones/glass[11,12], and carbon elastomers/carbon elastomers[13,14]. However, adhesions become difficult when the surface chemistry of the module varies significantly from that of the substrate (e.g., silicones vs. carbon elastomers). One substrate that can universally adhere to different modules and intimately deform with them remains unavailable. On the other end, spatially tailoring the stretchability of the substrate through composition and geometry designs has been introduced to enable mechanical compliance between the module and substrate. The strain-insensitive regions of the substrate hold the modules, while the strain-sensitive regions respond to the external stress. Successful examples

[1]Department of Materials Science and Engineering, Southern University of Science and Technology, Shenzhen 518055, China. [2]School of Materials Science and Engineering, Yancheng Institute of Technology, Yancheng 224051, China. [3]Shenzhen Key Laboratory of Soft Mechanics & Smart Manufacturing, Department of Mechanics and Aerospace Engineering, Southern University of Science and Technology, Shenzhen 518055, China. [4]Guangdong Provincial Key Laboratory of Sustainable Biomimetic Materials and Green Energy, Southern University of Science and Technology, Shenzhen 518055, China. [5]These authors contributed equally: Yan Shao, Jianfeng Yan. ✉e-mail: renfz@sustech.edu.cn; yuyh@sustech.edu.cn

of such substrates include polydimethylsiloxane (PDMS)[15–17], Ecoflex[18–20], styrene-ethylene/butylene-styrene (SEBS)[9,13], and polyurethane (PU)[21,22]. The dominant strategy for making area-dependent stretchability is to create rigid island regions in mother elastomers by introducing crosslinking or hard units, while softening mother elastomers without substantially altering their barrier properties remains a challenge[23–25]. From adhesion to stretchability designs, contemporary developments of the packaging substrate lack a standard option that can universally assemble different units with module-specific mechanical compliance.

In this work, we report a packaging substrate with module-specific stretchability and adhesiveness by sectionally tuning molecular mobility in the bulk via the plasticizing effect and enhancing the molecular polarity on the surface via an interposer layer. The regional modulus of the substrate can be finely controlled from 60 kPa to 3 MPa depending on the needs of assembly. The transition between different sections was seamless and sharp, preventing the substrate from structural failure. The interposer layer was made by introducing highly polar domains in an elastic polymer matrix, through which unifying the attributes of chemical bonding and topological entanglement. The substrate with the interposer layer formed robust yet stretchable adhesions with silicon (Si), aluminum (Al), poly(3,4-ethylenedioxythiophene):poly(styrene sulfonate) (PEDOT:PSS), PET (polyethylene terephthalate); PDMS, Ecoflex, SEBS, and hydrogel. The adhesion and stretchability design jointly result in the reliable integration of multiple modules that can endure arbitrary deformations. Exploiting these unique features of this substrate, we fabricated an arbitrarily deformable bioelectric generator/sensor and demonstrated its mechanical stability under stretching, compressing, bending, twisting, and crumpling. The intrinsically low permeability and high biocompatibility of the substrate further allowed the device to operate stably under in vivo conditions for 10 weeks.

## Results

### Design principle

Stretchable electronics often consist of mechanically mismatched components: stiff and brittle modules adhered onto soft and stretchable substrates. Subject to a stretch, the prominent discrepancy in modulus gives rise to severe stress concentration at the soft/stiff interface, which is the primary cause of interfacial delamination. From the perspective of fracture mechanics, interface delamination is driven by the energy release rate ($G$), which relates to the strain and modulus of the module and substrate through the following equation:

$$G = Z h_m \sigma^2 / E_m \qquad (1)$$

where $\sigma$ is the tensile stress and $\sigma = E_s \varepsilon$, $\varepsilon$ is the strain, $h_m$ is the thickness of the module, $E_m$ and $E_s$ are the Young's modulus of the module and substrate, respectively, $Z$ is a dimensionless driving force that determined by the cracking pattern and the modulus mismatch between the module and substrate[26]. This equation indicates that $G$ increases monotonically with the strain and the modulus mismatch between the module and substrate. Delamination occurs when $G$ exceeds the interfacial toughness $\Gamma$ (i.e., $G > \Gamma$)[27]. Therefore, improving the mechanical stability of the assembled modules on the substrate requires a low modulus mismatch for reducing $G$ and a high interfacial toughness for increasing $\Gamma$.

Directed by the above principle, we endowed the packaging substrate with module-specific stretchability to reduce the modulus mismatch and strong adhesion to increase the interfacial toughness. We chose poly(styrene-isobutylene-styrene) (SIBS) as the mother material for the substrate considering its high elasticity and chemical stability, and its lowest water permeability among elastomers[28]. The elastic modulus of SIBS can be broadly tuned by introducing polyisobutylene (PIB) oligomers, creating area-selective stretchability for a wide range of stretchable and stiff modules (Fig. 1a). The interposer layer is composed of SIBS and maleic anhydride grafted polypropylene (PP-g-MAH). The SIBS chains and MAH groups in the interposer layer are responsible for topologically entangled with the SIBS substrate and chemically bonded with the module surface, respectively.

We observed an order of magnitude improvement in the degree of strain that triggers the assembly failure for this new packaging substrate compared to conventional stretchable substrate. For a homogenous substrate without the interposer layer, both stiff and stretchable modules delaminated under a small strain (20%, case 1 in Fig. 1b, c). Module-specific design prevented the delamination of the stiff module, while the stretchable module delaminated under 100% tensile strain due to the weak adhesion (case 2 in Fig. 1b, c). Strong interfacial adhesion improved the structural stability of the stretchable module, but the stiff module still delaminated under 200% tensile strain due to the concentrated stress from the modulus mismatch between the module and substrate (case 3 in Fig. 1b, c). For the module-specific substrate with the interposer layer (i.e., a combination of bulk modulus and surface adhesion designs), both stiff and stretchable modules maintained their structural stability under 600% strain (case 4 in Fig. 1b, c), the largest strain degree for a module/substrate interface to the best of our knowledge (Table S1, Supplementary Information). For the case 4 in Fig. 1b, the lower $E_m$ of the soft module leads to a higher $G$ value relative to the hard module under the same strain according to the above equation. For the case 3 and 4 in Fig. 1b, the interposer layer mainly increased $\Gamma$. The interposer layer enables stronger interfacial adhesion (i.e., higher interfacial toughness $\Gamma$), so that the interface can tolerate a larger strain and a higher $G$. The finite element simulation confirms that the module-specific substrate greatly reduced the magnitude of the concentrated stress and thus mitigated the debonding issue. For example, when subject to a nominal strain of 200%, the maximum Mises stress of the integrated device based on a conventional homogenous substrate reached 12.46 MPa (case 3), which was 2.68 times that of the module-specific substrate in case 4 (4.65 MPa, Fig. 1d and Fig. S1). These results highlight the importance of combining the bulk and surface designs in the packaging substrate.

The high mechanical integrity leads to the steady operation of an actual operating circuit under large strains (Fig. S2 and Movie S1). The circuit includes stretchable conducting lines made from a SIBS/PIB/liquid metal composite and stiff modules such as light emitting diode (LED), inductor, chip, diode, and resistor. The stiff modules and stretchable conducting lines were integrated onto the hard and soft regions of the packing substrate, respectively. When the circuit was externally powered and stretched, the LED remained lit up under a stretch of 400%. Further increasing the stretch made the LED go out due to the deformation-induced electrical breakdown of the stretchable conducting lines.

**Module-specific stretchability.** The customizable stretchability was originated from plasticizing SIBS chains with PIB oligomers.

The SIBS/PIB blend exhibited different surface features compared to pristine SIBS under the phase scan of taping-mode atomic force microscopy (AFM). The distance between the PS blocks of SIBS was enlarged from 24 nm to 42 nm after adding PIB, implying the disentanglement of SIBS chains due to the plasticizing effect induced by the PIB oligomer (Figs. 2a and S3). The glass transition temperature ($T_g$) of SIBS was reduced from −34 °C to −39 °C after introducing 20 wt% of PIB, further evidencing the enhanced molecular mobility with the plasticizing effect of PIB oligomer[29,30]. Due to the low content of PS, $T_g$ of the PS segments was not observed on both samples in the tan δ spectra, but appeared in the SIBS with 30 wt% PS (Figs. 2b and S4), consistent with previous observations on the PS-based block copolymers using dynamic mechanical analysis[31,32]. However, the $T_g$ of PS for all samples was detectable on differential scanning calorimetry (Fig. S5).

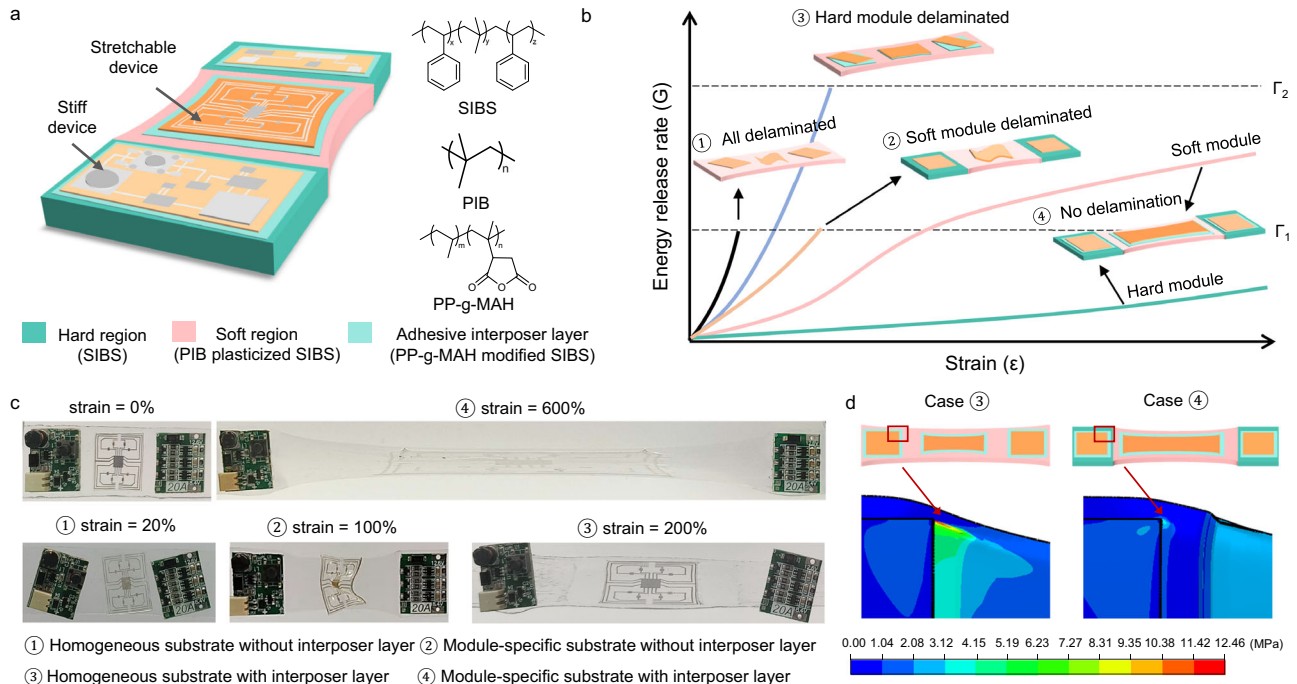

**Fig. 1 | Device assembly on packaging substrates. a** Schematic of the packaging substrate with module-specific stretchability and adhesiveness. The hard and soft regions were designed to hold stiff and stretchable devices, respectively. SIBS represents poly(styrene-isobutylene-styrene); PIB represents polyisobutylene; PP-g-MAH represents maleic anhydride grafted polypropylene. **b** Schematic plot for the crack driving force – energy release rate (*G*) as a function of total strain. *Γ₁* and *Γ₂* refer to the interfacial toughness between the module and substrate with and without the adhesive interposer layer, respectively. Case 1 to 4 refer to the modules integrated onto the homogeneous substrate without the interposer layer (1), onto the module-specific substrate without the interposer layer (2), the homogeneous substrate with the interposer layer (3), the module-specific substrate with the interposer layer (4). The hard and soft regions in case 4 were separately plotted. **c** Photographs for the experimental observation of the mechanical stability of the four situations depicted in (**b**). **d** Finite element simulation of the stress distribution for case 3 (left) and 4 (right), identifying concentrated stress at the module/substrate interface in case 3, but not in case 4. The scale range for these two cases was unified to 12.46 MPa for a clear comparison.

---

The elastic moduli of a pristine SIBS with 15 wt% of PS and the same SIBS modified by 5, 20, 40, and 60 wt% of PIB (molecular weight: 1300) were 840, 680, 420, 200 and 60 kPa, respectively (Figs. 2c and S6). The monotonic decrease of modulus with the increase of PIB content was a direct result of the enhanced chain mobility observed by the AFM and $T_g$ characterizations. The elongations at break of the pristine SIBS and the SIBS with 5, 20, 40, and 60 wt% of PIB were 680%, 850%, 800%, 640%, and 430%, respectively. The initial increase of elongation was caused by PIB-induced disentanglements of SIBS chains. When the PIB content exceeded 40%, the elongation at break decreased compared with the pristine SIBS due to the shorter chain length of the dominant PIB oligomers.

The SIBS films with varied modulus can be integrated into one substrate to yield regional stretchability by a solvent welding process. As a demonstration, we chose a SIBS film containing 30 wt% of PS as the hard region and a SIBS containing 15 wt% of PS and 20 wt% of PIB oligomers as the soft region. For both the soft-hard-soft (SHS) and hard-soft-hard (HSH) films, the deformation mainly occurred in the soft region when the stress was low (Fig. 2d). The stress-strain curves showed a similar mechanical behavior determined by the soft region for the soft film, SHS film, and HSH film when the strain was <100% (inset in Fig. 2e). With the increase of stress and strain, the hard regions experienced the load and differentiated the mechanical behaviors of these films (Fig. 2e). The elongation at break gradually decreased with the increase of the portion of hard regions since the hard region had fewer stretchable segments. The surface of the transition area between soft and hard regions was clean (Fig. S7a). The physical size of the transition area (i.e., the resolution) depends on the solvent evaporating time of the hard region, which controls the solidification degree of the hard region (Fig. S7b–f). The smallest transition distance between the soft and hard regions was about 100 μm.

The enhanced chain mobility increased the probability for water to penetrate, causing a side effect of the enlarged water vapor transmission rate (WVTR). The WVTR of the pristine SIBS was measured to be 0.35 g·mm·m⁻²·day⁻¹ at 38 °C and 90% relative humidity (RH, Fig. S8). For the SIBS film with <40 wt% of PIB, the increase of WVTR was marginal (<20%). The WVTR of the SIBS film with 60 wt% of PIB abruptly increased to 0.52 g·mm·m⁻²·day⁻¹, due to the new free volume created by the dominant PIB oligomer[24,33]. The WVTR of the SHS and HSH films were both 0.35 g·mm·m⁻²·day⁻¹ at 38 °C and 90% RH, similar to that of the soft and the hard SIBS films (Fig. 2f). This result indicated that the interfaces between the soft and hard regions were non-defective, consistent with the cross-section SEM observation (inset in Fig. 2f).

The WVTR of SIBS is orders-of-magnitude lower than that of commonly used elastomers, such as SEBS (~5 g·mm·m⁻²·day⁻¹) and PDMS (~70 g·mm·m⁻²·day⁻¹). The low WVTR of SIBS originated from its molecular composition and structure. SIBS is a triblock copolymer composed of a PIB elastic block and PS end-blocks. The fully saturated nonpolar C-C backbone provides strong chemical repellence to the polar water molecule. The dense methyl side groups in the PIB segments create a large steric hindrance, further preventing the penetration of water. Compared to SIBS, the Si-O bond in silicones has higher tendency to attract water molecules due to its polarity[34] and the steric hindrance effect of the ethylene–butadiene segment in SEBS is weaker than that of the PIB segment[35]. The low WVTR of SIBS suggests they can more effectively protect devices from moisture-induced damages relative to SEBS and PDMS.

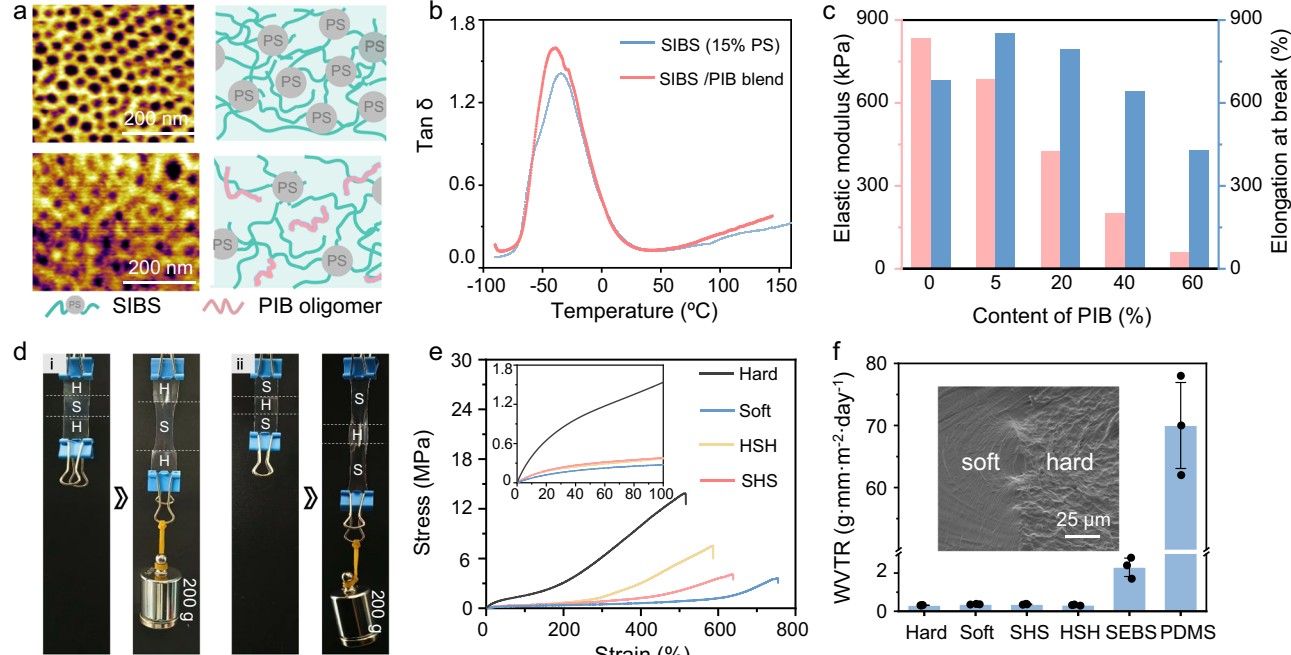

**Fig. 2 | Structure and property of the packaging substrate with module-specific stretchability. a** AFM phase images and structural schematic of pristine SIBS (top) and the SIBS blended with 20% of PIB (bottom). Representative results are presented from *n* = 5 independently repeated experiments. **b** Tan δ spectra of pristine SIBS and SIBS/PIB blend films. The glass transition temperature shifted from −34 °C to −39 °C after introducing PIB, suggesting the enhanced chain mobility of SIBS. **c** Elastic modulus and elongation at break as a function of the PIB content for SIBS/PIB blend films. **d** Photographs of hard-soft-hard (HSH, i) and soft-hard-soft (SHS, ii) SIBS films before and after being pulled by a 200 g of weight, showing that deformations mainly occurred in the soft regions. H and S represent the hard and soft regions, respectively. **e** Stress-strain curves of hard, soft, HSH, and SHS films. The inset is a magnified illustration for the region within 100% of strain. Hard and soft refer to the hard SIBS and the soft SIBS, respectively; HSH and SHS refer to the hard-soft-hard SIBS and the soft-hard-soft SIBS, respectively. **f** Water vapor transmission rate of hard, soft, HSH, SHS, SEBS, and PDMS films measured at 38 °C and 90% relative humidity. The error bars represent standard deviations and data is from *n* = 3 biologically independent samples and expressed as mean ± s.d. SEBS refers to styrene-ethylene/butylene-styrene and PDMS refers to polydimethylsiloxane; The inset is a cross-section SEM image that shows the seamless transition between the soft region and hard region in HSH and SHS films.

**Adhesion induced by the interposer layer.** We made the substrate adhesive to different materials by introducing an interposer layer composed of SIBS and PP-g-MAH. The interposer layer and the substrate can easily adhere with each other through topological entanglement of SIBS chains. On the other direction, the MAH group has a strong polarity and can form covalent bonds with amine groups, allowing the interposer to chemically bind with surfaces treated by aminosilane molecules like (3-aminopropyl) triethoxysilane (APTES, Fig. 3a). Through a fairly facile blade casting (Fig. S9) and warm pressing process (80 °C and 200 g of weight), conformable adhesions were formed between the substrate and several commonly used materials in stretchable electronics, including PDMS, Al foil, Si, PEDOT: PSS, PET, and chitosan hydrogel (Fig. 3b). The interfacial toughness for PET, Al, PDMS, and Si reached 390, 220, 200, and 120 J·m⁻², respectively, about one order of magnitude higher than those of the interfaces without the interposer layer. The adhesion with an interfacial toughness of above 100 J·m⁻² is sufficient for most applications, such as bioelectronic devices, stretchable conductors, and stretchable circuits[27,36–38]. The interfacial toughness for PEDOT: PSS and chitosan hydrogel was unmeasurable since they exceeded the cohesive energy of the material, i.e., the material ruptured before the interface was separated (Fig. S10). Significantly higher interfacial toughness (>900 J·m⁻²) could be obtained by pressing these interfaces at 150 °C using laminating equipment (Fig. S11). This is the first stretchable packaging substrate that can universally adhere metals, ceramics, plastics, and elastomers with distinct surface chemistries to the best of our knowledge (Table S2).

To quantitatively evaluate the influence of the regionally designed stretchability to the adhesion stability, we measured the variation of interfacial toughness of the APTES-treated Al film adhered on the hard region of the HSH substrate as a function of the substrate deformation and compared it with the same adhesion on a homogeneous SIBS substrate (Figs. 3d and S11). The interfacial toughness of Al/HSH interface was nearly independent of the substrate deformation within 600% of strain since the soft region dissipated most of the stress and the hard region remained intact. In contrast, the Al film delaminated from the homogeneous SIBS substrate when the substrate was stretched to 100%, leading to a dramatic decrease of the interfacial toughness. The Al/HSH interface maintained its interfacial toughness after undergoing 20,000 cycles of 100% tensile strain, demonstrating its working stability during long-term operation (Figs. 3e and S12).

**Arbitrarily deformable bioelectronic device assembled from the packaging substrate.** The integration of modules was demonstrated through assembling a fully stretchable bioelectronic generator/sensor using the HSH and SHS packaging substrates. Multiple domains of PDMS and SIBS-protected hydrogel were attached to the hard region of the packaging substrates to serve as dielectrics and electrodes, respectively (Figs. 4a and S13). The working principle of this device was based on triboelectric and electrostatic effects through the following process. (i) At the starting position, the top and bottom modules were fully overlapped and intimately in contact with each other. Positive and negative charges accumulated on the SIBS and PDMS surfaces after triboelectrification, respectively, due to their different abilities to hold electrons (Fig. S14). (ii) When the device was stretched, a relative displacement between the top and bottom charging layers was produced as a result of the region-specific deformation of the SHS and HSH substrates. The reduced overlapping area changed the electrostatic

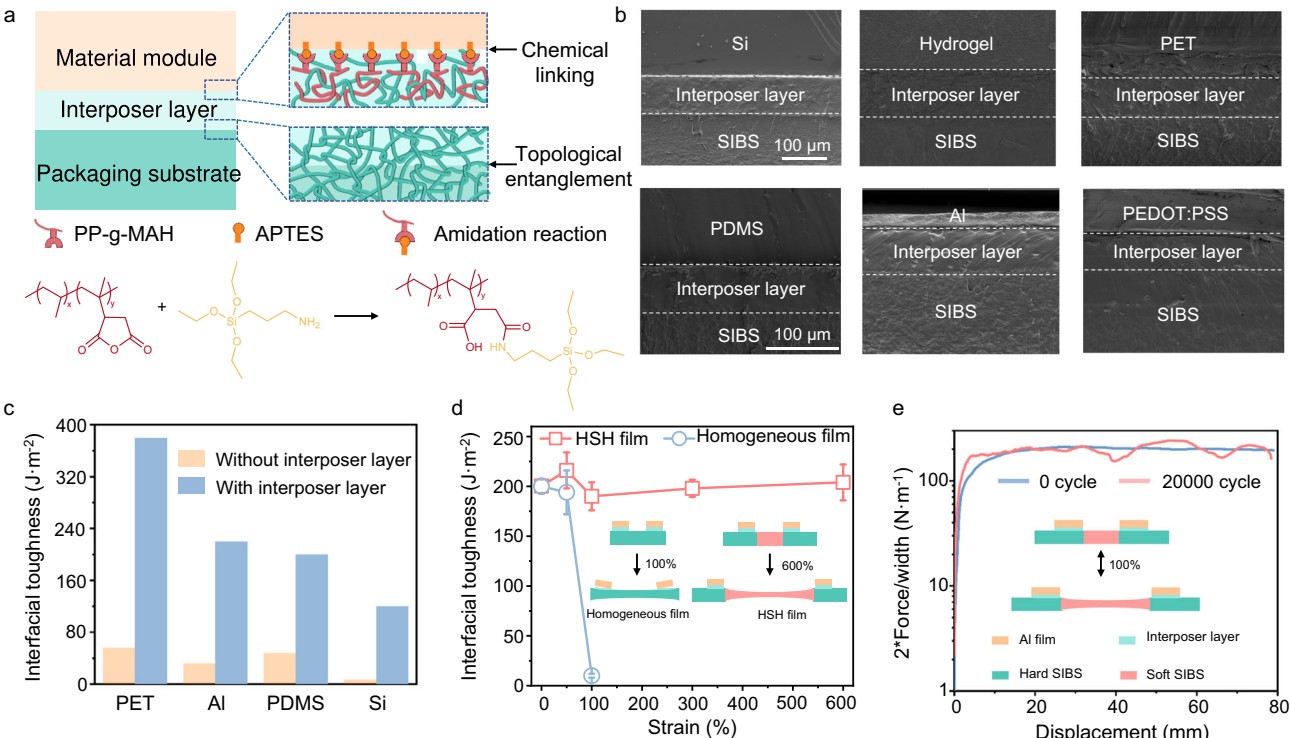

**Fig. 3 | Adhesion between the packaging substrate and different materials. a** Adhesion mechanism between the packaging substrate and material module by introducing an interposer layer between them. The reaction shown at the bottom refers to chemical linking between the PP-g-MAH molecules in the interposer layer with the APTES-treated module surface. APTES represents (3-aminopropyl) triethoxysilane. **b** SEM images of the tightly adhered interfaces between the packaging substrate and Si (silicon), hydrogel, PET (polyethylene terephthalate), PDMS, Al (aluminum), and PEDOT:PSS (poly(3,4-ethylenedioxythiophene):poly(styrene sulfonate)) formed through the interposer layer. Representative results are presented from n = 3 independently repeated experiments. **c** Interfacial toughness comparison of the packaging substrate with and without interposer layer on PET, Al, PDMS, and Si. **d** Interfacial toughness of an Al/HSH interface and an Al/homogeneous SIBS interface after being stretched to 50%, 100%, 300%, and 600%. The left inset is a schematic diagram showing that Al delaminated from the homogeneous SIBS film after undergoing 100% tensile strain. The right schematic illustrates that Al kept stable on the HSH film after undergoing 600% tensile strain. The error bars represent the standard deviation and data is from n = 3 biologically independent samples and expressed as mean ± s.d. **e** Force/width - displacement curves of Al/HSH before and after stretched to 100% for 20000 cycles.

equilibrium, driving the electron flow from the reference electrode (a ground electrode) to the hydrogel electrode and generating an electrical current. (iii) The current maximized at the position when the top and bottom charging layers were fully separated. (iv) With the back movement, electrons on the hydrogel electrode flew back to the ground to keep the electrostatic equilibrium, generating an opposite current. Alternative current pulses were continuously produced by repeating this stretch-release cycle (Fig. 4b and Movie S2).

When using this device as a generator that converts mechanical energy to electricity, its voltage output reached a peak-to-peak value ($V_{pp}$) of ~7 V (Fig. 4c). In comparison, the voltage outputs of the generators assembled using a homogeneous soft substrate or hard substrate were orders of magnitude lower under the same stress due to the small relative displacement of the charging layer. The voltage and current outputs of this device increased monotonically with the increase of strain, ranging from 0.5 V and 4 nA for the strain of 10% to 7 V and 16 nA for the strain of 150% (Figs. 4d, e, and S15). This is an intuitive result since the contact and separation area is proportional to the strain.

This device can convert arbitrary mechanical stimuli to electrical signals due to its high mechanical and electrical sensitivities endowed by the SHS and HSH designs. The contact-separation processes of the SHS and HSH substrates differ dramatically under bending, twisting, crumpling, and compressing (Fig. S16), leading to distinctive peak shapes and magnitudes in the current signals (Figs. 4f and S17).

The superior structural stability and low WVTR of the packaging substrates endow the device with high mechanical and chemical stabilities. Its outputs remained steady being immersed in acidic and alkaline phosphate buffer saline (PBS) solutions for 5 weeks (Figs. 4g and S15). After 15,000 cycles of continuous deformation to 70%, the voltage output of the device remained at ~3 V (Fig. 4h). The sensitive response to arbitrary deformations, together with the high mechanical and chemical stabilities, makes this device a versatile system for sensing body motions or converting biomechanical energy to electricity.

**In vivo operation of the bioelectronic device.** The main component of this packaging substrate—SIBS is an implantable material approved by the Food and Drug Administration (FDA) of the United States[39]. Whether the polar MAH groups are biosafe or not is a question under debating[40,41], but the interposer layer that contains MAH groups was placed at the inner surface of our packaging substrate and was physically separated with the body environment by pristine SIBS. Therefore, this packaging substrate and the device assembled from it will be fully biocompatible.

The biocompatibility of the packaging substrate was evaluated by incubating and comparing mouse fibroblast cells. The immunofluorescence staining images showed that the cells incubated with the HSH packaging substrate exhibited a typical elongated morphology with reasonable distributions and densities, similar to the cells in the blank group (Fig. 5a and Fig. S18). The live/dead staining assay revealed that cell viability on the HSH film was comparable to that on the blank surface (Fig. 5b). These findings provide evidence of the non-toxic nature of the packaging substrate.

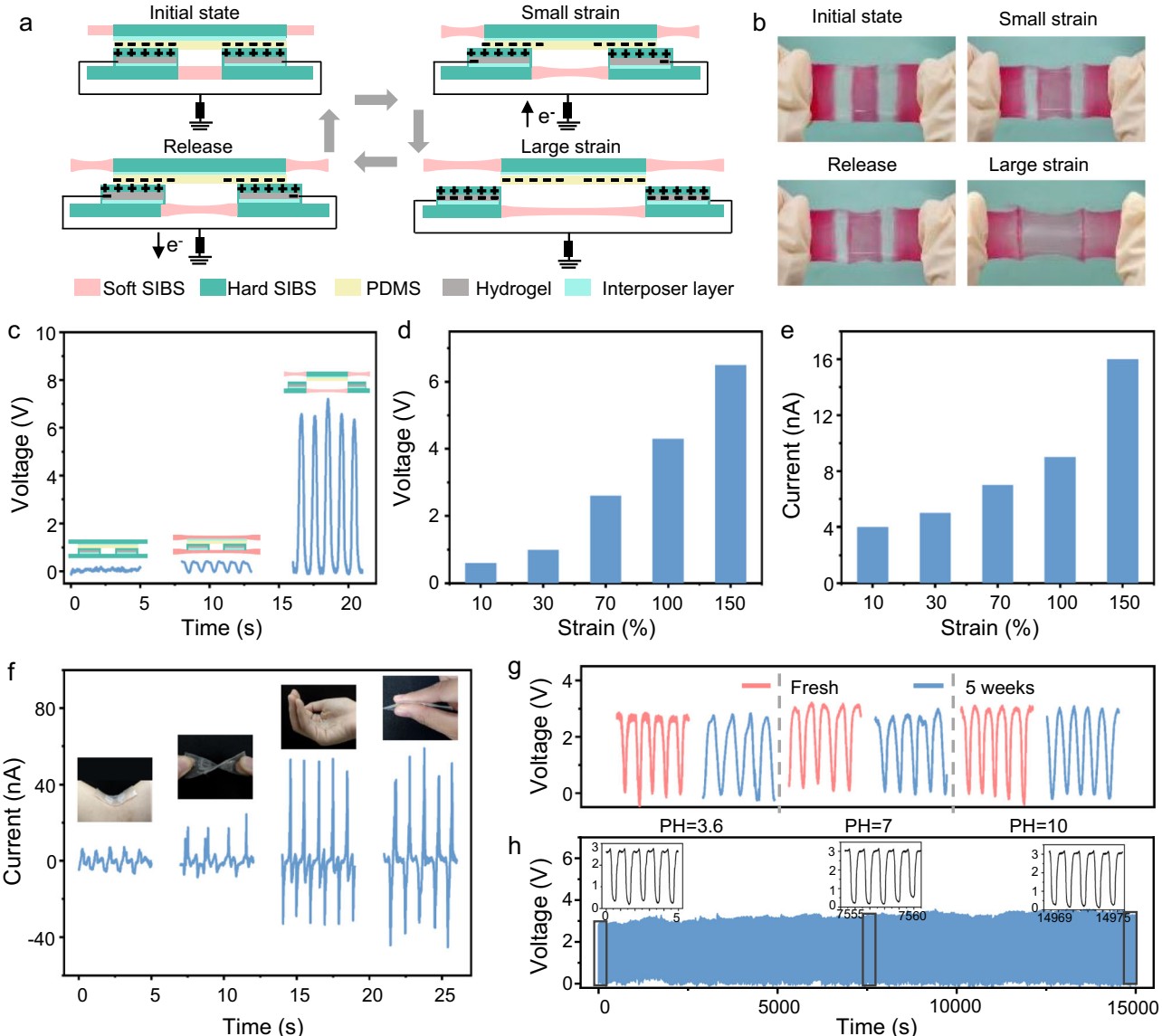

**Fig. 4 | Working principle and electrical outputs of the arbitrarily deformable bioelectronic device. a** Working principle of the device converting mechanical deformation to electrical signal. **b** Photographs of the moving process of the device shown in (**a**). The soft regions were dyed in red and the hard regions were kept transparent. **c** Voltage outputs of the devices assembled using a homogeneous hard substrate, a homogeneous soft substrate, and the module-specific substrate. Insets are the structural schematic of these three devices. **d**, **e** Voltage (**d**) and current (**e**) outputs of the device shown in (**a**) at different strains. **f** Current outputs of the device under different deformations, including bending, twisting, crumpling, and compressing. The insert photographs show the deformation modes when recording the signals. **g** Voltage outputs of the device before and after being immersed in PBS solution with different PH values for 5 weeks. **h** Output stability of the device under 15, 000 cycles of stretching to 70%. The insets show the voltage output of the device at the beginning, middle, and end of the test.

Accordingly, the bioelectronic device assembled using these packaging substrates (Fig. 4a) exhibited high compatibility with the tissue. Figures 5c and S19 show the pathological analyses after 2 and 10 weeks of implantation, respectively. No inflammation of lymphocytes, tissue injury, or morphological change in muscular cells was observed in the top and bottom tissues surrounding the device. Neither tissue injury nor morphological change was observed in muscular cells. No deformation or abnormal lymphatic cell invasion was found in vital organs, including the heart, lung, liver, spleen, and kidney. Blood and serum analyses revealed comparable values in white blood cells (WBC), neutrophils, and lymphocytes for both the experimental and control groups, showing no signs of inflammation. Hematopoietic function, as indicated by red blood cell (RBC), remained stable during the testing period, showing no signs of anemia (Fig. 5d).

The device operated stably when implanted under the skin of adult rats in the dorsal, thigh, and chest (Fig. 5e and Fig. S20). The device placed on the chest generated a stable voltage of 0.5 V in response to the rat's respiration (Fig. 5f and Movie S3), demonstrating its ability to sense subtle biomechanical signals. The devices placed in the thigh and dorsal regions produced a consistent voltage of 0.7 V and 1.5 V after gently stretching the dorsal region and the leg at a frequency of 1 Hz, respectively (Movie S4, S5). The voltage output of the implanted device remained unchanged at 1.5 V for 10 weeks, making the longest in vivo lifetime among the stretchable implantable biomechanical sensor to the best of our knowledge (Table S3, Supplementary Information). This result stemmed from the low WVTR and module-specific stretchability and adhesivity of the packaging substrate (Figs. 2f and S21, S22), concurrently preventing the device from

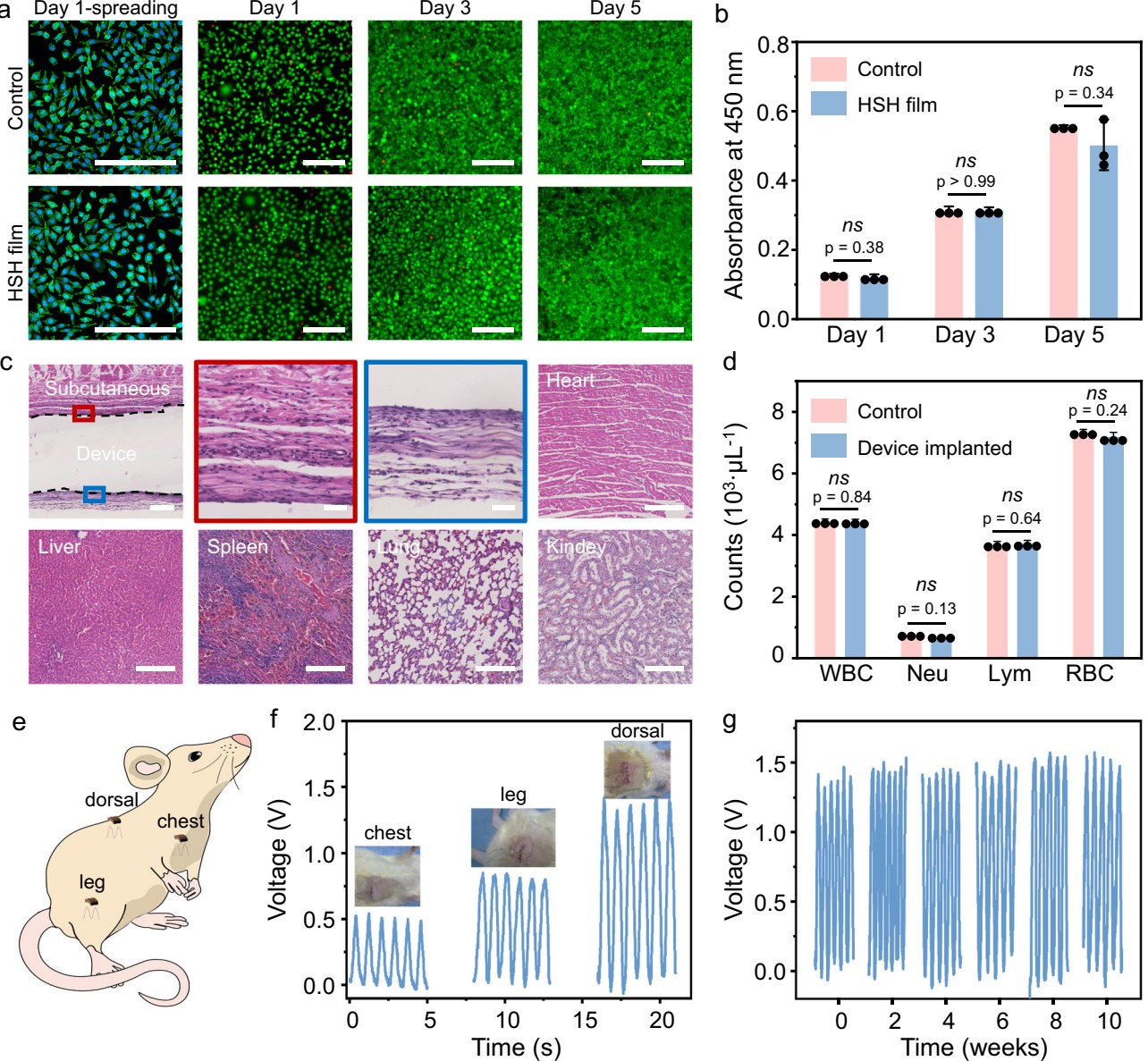

**Fig. 5 | Biocompatibility and in vivo performance of the arbitrarily deformable bioelectronic device. a** Cytotoxicity comparison between control and HSH film. Column 1: Fluorescence micrograph showing the normal morphology of L929 cells over a 1 day culturing period. Green indicates the cytoskeleton, and blue indicates the nucleus. Columns 2–4: Fluorescence micrographs of live/dead staining assay showing the growth of L929 cells during a 5 day culturing period. **b** Optical density at 450 nm of L929 cells growth during a 5 day culturing period. **c** Pathological analyses by H&E staining of regional tissues surrounding the implanted device and on most vital organs (heart, liver, spleen, lung, and kidney) after 2 weeks of implantation, revealing no signs of inflammation. **d** Blood tests comparison with and without device implantation during a 2 week implantation period, including white blood cells (WBC), neutrophils (Neu), lymphocytes (Lym), and red blood cells

(RBC). All scale bars are 200 μm. **e** Schematic illustration showing the positions of device implantation, including the dorsal, thigh, and chest areas. The device function as a sensor when implanted on the chest and as a generator when implanted on the dorsal and thigh areas. **f** In vivo electrical signals of the device produced by natural respiration when implanted on the chest (left) and generated by gentle stretching when implanted on the thigh (middle) and dorsal region (right). Insets are the surgery photographs of the device implantation. **g** In vivo electrical signals produced by the device during continuous implantation on the dorsal region for 10 weeks, recorded at 2 week intervals. Note: Data in (**b**) and (**d**) are from $n = 3$ biologically independent samples and expressed as mean ± s.d. Statistical differences were analyzed with two-sided $t$-test, *$p < 0.05$. No data were excluded from the analyses.

chemical failure induced by body fluids and mechanical failure induced by deformation.

## Discussion

In summary, we present a packaging substrate that offers module-specific stretchability and adhesiveness, low water permeability, high chemical stability, and high biocompatibility. The substrate was achieved by modifying SIBS with PIB plasticizers and an adhesive interposer layer. The modulus of the substrate can be finely adjusted

from 60 kPa to 3 MPa in different regions based on the assembly requirements. Adhesion can be universally formed on the surfaces of metals, ceramics, plastics, and elastomers under mild conditions. The regional stretchability and tight adhesion together ensure stability at the module/substrate interface under 600% tensile strain. The WVTR of the substrate was 0.35 g mm m$^{-2}$ day$^{-1}$ at 38 °C and 90% RH, orders of magnitude lower than that of conventional SEBS and PDMS substrates. As proof of concept, we integrated conductive and dielectric material modules with the packaging substrate, creating a highly

stretchable bioelectronic device capable of generating electricity from arbitrary deformations. Due to the unique combination of favorable mechanical, chemical, and biomedical properties, the device exhibited high operational stability under 15,000 cycles of 70% tensile strain, 5 weeks of immersion in PBS solution, and 10 weeks of implantation in a living rat. This packaging substrate stands as a versatile choice for assembling stretchable electronics designed to operate in complex mechanical and physicochemical conditions.

## Methods

### Fabrication of the packaging substrate with regional stretchability

The fabrication started with preparation of SIBS/PIB blend films. 5 g of SIBS (15 wt% of PS) and 1.25 g of PIB were dissolved in 20 mL of toluene and stirred for 4 h at room temperature to obtain a homogeneous precursor solution. 9 mL of the precursor was casted on a rectangular container with a size of $8 \times 6.5\ cm^2$. A transparent SIBS/PIB blend film with a thickness of $250\ \mu m$ was obtained after the solvent was evaporated at room temperature in the fume hood. SIBS (15 wt% of PS)/PIB (8:2, wt/wt) and pristine SIBS with 30 wt% of PS were selected as the soft and hard components for the hard-soft-hard (HSH) and soft-hard-soft (SHS) films. For the HSH film, 3.4 mL of 250 mg/mL SIBS (30 wt% PS)/toluene solution was respectively dropped in the left ($3 \times 6.5\ cm^2$) and right part ($3 \times 6.5\ cm^2$) of a rectangular container with a size of $8 \times 6.5\ cm^2$. A PDMS film ($2 \times 6.5\ cm^2$) was inserted in the middle to leave the space for the soft component. The SIBS films in the left and right parts of the container were controlled to be semi-solidified by partially evaporating toluene at room temperature in the fume hood. After that, removing the PDMS spacer and filling the blank space with 2.2 mL of SIBS/PIB/toluene solution. Due to the diffusion of toluene during the film solidification, the edges between the soft part and hard part were seamlessly merged. At last, a transparent HSH film was obtained after the solvent was fully evaporated at room temperature in the fume hood. The SHS film was fabricated via a similar process. The dimension of each part can be controlled by changing the size of the mold and the amount of the precursor solution.

### Adhesion of the packaging substrate with different materials

Preparation of PP-g-MAH/SIBS solution: 2 g of PP-g-MAH was dissolved in 100 mL of toluene and stirred for 2 h at 120 °C in a 500 mL three-necked flask equipped with a condenser, a thermometer, and a mechanical stirrer. After that, 40 g of SIBS (15 wt% of PS) was added to the toluene solution and stirred for 5 h at 80 °C to obtain the PP-g-MAH/SIBS precursor solution.

Adhesion with plastics, semiconductor, and metals: PET, Si, Al foils were first cleaned with deionized water and isopropanol, followed by oxygen plasma treatment for 5 min at 300 W. These surfaces were subsequently immersed in an ethanol/water (volume ratio 9:1) solution containing 5 wt% of APTES for 2 h, followed by drying at 80 °C for 1 h. The interposer layer with a film thickness of about $100\ \mu m$ was produced by blade-coating the PP-g-MAH/SIBS precursor solution on APTES-treated surfaces. The adhesions were formed by pressing the packaging substrate onto the treated surfaces with 200 g weight in an 80 °C oven for 4 h.

Adhesion with elastomer: The elastic PDMS films were prepared with a silicone base to curing agent ratio of 10:1. The mixture were manually stirred with a glass rod for 10 min, followed by degassing in a vacuum chamber for 10 min. 3 mL of PDMS precursor solution was dropped in a rectangular container with a size of $5 \times 5\ cm^2$. A PDMS film with a thickness of ~1 mm was obtained after curing the precursor in an 80 °C oven for 4 h. The PDMS film was treated under oxygen plasma at 300 W for 5 min and then immersed in an ethanol/water (volume ratio 9:1) solution containing 5 wt% of APTES for 2 h, followed by drying at 80 °C for 1 h. The interposer layer with a film thickness of $100\ \mu m$ was produced by blade-coating the PP-g-MAH/SIBS precursor solution on

APTES-treated surfaces. The adhesions were formed by pressing the packaging substrate onto the treated surfaces with 200 g weight in an 80 °C oven for 4 h.

Adhesion with PEDOT:PSS: The original PEDOT:PSS solution (PH1000) was filtered with $0.22\ \mu m$ syringe filters and then mixed with 5 vol% DMSO. The resultant solution was sonicated for 10 min to obtain a uniform mixture. Free-standing PEDOT:PSS films were fabricated via a simple solution casting process, followed by drying at 80 °C for 1 h. The films were treated by oxygen plasma at 300 W for 5 min. Then, the films were immersed in an ethanol/water (volume ratio 9:1) solution containing 5 wt% of APTES for 2 h, followed by drying at 80 °C for 1 h. After that, PP-g-MAH/SIBS solution was bladed coating on APTES treated surface to form adhesive interposer layer. The adhesions were formed by pressing the packaging substrate onto it with 200 g weight in the oven with the temperature of 80 °C for 4 h.

Adhesion with chitosan hydrogel: 1 g of chitosan power was dissolved in a solution mixture containing 40 mL of DI water, 10 mL of glycerol, and 2 mL of acetic acid under stirring at room temperature for 4 h. After degassing, 20 mL of chitosan precursor solution was casted into a petri dish with a diameter of 8 cm, and evaporated in a 40 °C oven for 12 h to obtain the chitosan gel. The adhesion between chitosan gel and SIBS film was form by placing the PP-g-MAH/SIBS/toluene solution in between and evaporating the toluene naturally. Conductive chitosan hydrogel was achieved by immersing the prepared chitosan hydrogel in to 5% NaCl/ DI water for 5 h.

Finite element simulation: The static finite element analysis was implemented in Abaqus/Standard. The model was in a 1:1 scale according to the actual sample (the simulation area in Fig. S1a). The soft region (pink) and stiff segment (green) of the substrate, and the soft modules (orange) were modeled as neo-Hookean materials with elastic moduli of 400 kPa, 3 MPa, and 400 kPa, respectively. The stiff modules (yellow) were modeled as elastic materials with elastic modulus of 1 GPa. All components were merged together without interface failures. Hybrid hexahedral structured elements (C3D8H) were used and the global mesh size was set to be 0.15 mm, resulting in a total of 304920 elements. The tensile load was applied to left and right surfaces of the stretchable device such that the nominal strain of the middle segment is 200%.

### Assembly of the arbitrarily deformable bioelectronic device

The SIBS, PDMS and conductive chitosan hydrogel films were used as the triboelectric layers and electrode to fabricate the arbitrarily deformable device. For the top part of the device, a PDMS film ($3 \times 1.8\ cm^2$, $100\ \mu m$ in thickness) with adhesive interposer layer was attached to the hard region of the SHS film ($1 \times 2.8\ cm^2$ for the soft region and $3 \times 2.8\ cm^2$ for the hard region, $250\ \mu m$ in thickness) by hot-pressing at 150 °C. For the bottom part of the device, two pieces of chitosan hydrogel ($1.2 \times 1.8\ cm^2$, 1 mm in thickness) with the adhesive interposer layer was adhered to the hard region of HSH film ($1 \times 2.8\ cm^2$ for the soft region and $2 \times 2.8\ cm^2$ for the hard region, $250\ \mu m$ in thickness). Then, $100\ \mu L$ of SIBS/toluene solution was coated on the surface of the hydrogel and a SIBS triboelectric layer ($100\ \mu m$ in thickness) was formed after toluene was evaporated. The stretchable device was assembled by hot-pressing the HSH and SHS modules along the edges with the PDMS layer and SIBS coated hydrogel facing each other at 130 °C.

### Structural and property characterizations

Scanning electron microscopy (SEM): SEM images were acquired on SE, TESCAN MIRA3 with a 5 kV accelerating voltage. Samples were coated with Pt or Au using sputtering prior to the characterization.

Atomic force microscopy (AFM): AFM images were obtained using an Asylum Research MFP-3D Stand Alone scanning probe microscope operated in tapping mode. AC240TS cantilevers (Olympus, typical spring constant: 2 N·m$^{-1}$) were used to record AFM images with sizes of

1 μm² at a resonance frequency of 70 kHz. The set point was set 600 mV. The driving amplitude of AFM tip was set to 280 mV for the phase image in Fig. 2a. Under a lower driving amplitude (150 mV), the phase image of the pristine SIBS became vague, while the phase image of the SIBS/PIB blend remained clear.

Dynamic mechanical analysis (DMA): DMA was performed on a dynamic mechanical analyzer (TA-Q-800). The films were fixed on a tension clamp and strained to 0.1% at a frequency of 1 Hz, where the temperature was swept from −90 °C to 150 °C with a heating rate of 5 °C/min.

Measurement of stress-strain curves: The tensile properties were characterized by a mechanical testing machine using a rectangular geometry (CMT 6203, MTS SYSTEMS, China). Samples were made in a rectangular shape with a size of $50 \times 7$ mm² and a thickness of 250 μm. The tensile properties were characterized at room temperature with a strain rate of 30 mm/min. Three tests were conducted for each sample and the error bars represent the standard deviation ($n = 3$). Cyclic stretching measurement was carried out on the HSH film with a stretching velocity of 30 mm/min and a strain of 100%.

Measurement of water vapor transmission rate (WVTR): The WVTR of SIBS, SIBS/PIB blend film, HSH film, SHS film and SEBS were acquired by AQUATRAN Model 3 (AMETEK MOCON) at 38 °C and 90% relative humidity. The chamber area was 5.64 cm² and all the films has a thickness of 250 μm. The WVTR for PDMS exceeded the largest measurable value of the AQUATRAN Model 3 equipment, so the gravimetric method was used to determine it WVTR. In specific, the PDMS films were placed over the mouth of vials prefilled with anhydrous copper sulfate. The edges were sealed by a paraffin film to the mouth of vials, and further tightened by a hollow lid. All the vials have a diameter of 1.8 cm. After applying the film, the vial was placed in a beaker filled with DI water at 38 °C for 3 weeks. The vial was then removed from water and its weight increment was measured immediately. The WVTR was calculated using the equation:

$$WVTR = \Delta m/A/t/d \qquad (2)$$

where $\Delta m$ was the weight increment of the vial (g), $A$ was the exposed film surface area (m²), $t$ was the total time being immersed in water (day), and d was the thickness of PDMS. Three tests were conducted for all films. The mean value was reported, and the error bars represent the standard deviation ($n = 3$).

Measurement of interfacial toughness: The interfacial toughness was measured with an Electromechanical Universal Testing Machine (Model CMT6203) equipped with 100 N load cells at a tensile speed of 30 mm·min⁻¹. Samples were cut into a rectangle shape ($1 \times 10$ cm²). The measurement was performed by a 180° peeling test and the interfacial toughness was calculated through dividing the peeling force by the sample width (multiply by 2 for the T-shape measurement). For each sample, the toughness was averaged from at least three replicates. The adhesion stability upon cyclic mechanical deformation was acquired by measuring the interfacial toughness of Al/interposer layer/HSH sample (Al film adhered on the hard region of the HSH film) before and after being stretched with a linear motor to 100% strain for 20,000 cycles. The adhesion stability was also characterized by measuring the interfacial toughness after the Al/interposer layer/SIBS and Al/interposer layer/HSH samples being stretched to the strain of 50%, 100%, 300%, 600%.

## Electrical measurement of the arbitrarily deformable bioelectronic device

Output measurement: The output of the arbitrarily deformable device and the control devices were measured under the same stretching force. All the devices had the same size ($5 \times 2.5$ cm²). During the measurement, the front tip of the device was fixed on a stationary stage and the back tip was attached to a moveable stage, which was connected to a computer-controlled linear motor (P01-37 × 120-C_C1, Linmot). The devices were stretched and released periodically at a frequency of 1 Hz. The generated voltage, current and charge was recorded by the electrometer (Keithley 6514).

Stability evaluation: The output stability of the arbitrarily deformable device was characterized by measuring its outputs in PBS solution with the PH value of 3.6, 7 and 10. The entire device was completely immersed in PBS solution for 5 weeks and the output voltage was measured by an electrometer (Keithley 6514) once time a week. The device was stretched and released periodically with the strain of 70% at a frequency of 1 Hz and the output voltage was recorded at the same time.

## Biocompatibility and in vivo output of the arbitrarily deformable bioelectronic device

Cell cytotoxicity: L929 mouse fibroblast cells were purchased from BeNa Culture Collection Co., Ltd., and incubated according to the program given in the manual. CCK-8 solution, Calcein AM, Propidium Iodide and phosphate-buffered saline (PBS) solution were purchased from Shanghai Beyotime Biotechnology Co., Ltd. Eagle's minimum essential medium (EMEM), fetal bovine serum (FBS), penicillin-streptomycin solution (P/S) were purchased from Thermo Fisher Scientific Inc. The cell cultural medium was obtained by adding FBS and P/S to EMEM with a volume ratio of 1: 0.1: 9. The cell cytotoxicity of HSH films were assessed with a CCK-8 assay using L929 mouse fibroblast cells. Before cell seeding, UV sterilized samples of HSH films (diameter: 10 mm, thickness: 250 μm) were placed at the bottom of 48-well plates as the substrate for cell growth. No material was added to the control wells. There were three wells in each group. $2 \times 10^4$ cells were seeded into each well containing 500 μl cell culture medium. After incubation for 1 day, 3 days and 5 days at 37°C, 5% CO₂ and 95% air, cell culture medium was aspirated, 10 vol% CCK-8 solution was added to each well followed by incubation at room temperature for 1 h. Lastly, the cell culture medium in the wells was transferred one by one to a 96-well plate, and their optical density (OD) values at 450 nm were measured by a microplate reader (BioTek Instruments, Inc.).

Cell Morphology Examination: UV sterilized samples of HSH films were placed at the bottom of 48-well plates as the substrate for cell growth. No material was added to the control wells. There were three wells in each group. $2 \times 10^4$ cells were seeded into each well containing 500 μl cell culture medium. After 24 h, the cytoskeleton and nucleus of the cells were stained with FICT Phalloidin (YESEN, 40735ES75) and DAPI (Beyotime, C1002), respectively. The staining protocol includes following procedures: First, the cell culture medium was discarded, and cells were washed two times with PBS. They were later fixed with 4% paraformaldehyde for 15 min and rinsed three times with PBS. The cell samples were then permeabilized with 0.1% Triton X-100 for 15 min at room temperature and rinsed three times with PBS. The samples were incubated for 30 min at room temperature with 200 μL/well of Alexa Fluor 555 Phalloidin working solution and then washed three times with PBS. Finally, fluorescent mounting medium with DAPI was added and incubated for 5 min at room temperature before applying cover slip. After staining, the cells were imaged using a Leica TCS SP8 X white light laser confocal microscope.

Cell live/dead staining assay: The experimental procedure of cell live/dead staining assay was the same as that of the CCK-8 assay before the replacement of the old cell culture medium. In the staining assay, the cell culture medium of each well. They were later filled with cell live/dead staining solution (200 μL staining solution each well), followed by re-incubation for 15 min to stain the cells. The staining solution was made by adding Calcein AM (staining live cells, fluorescence excitation at 494 nm) and Propidium Iodide (staining dead cells, fluorescence excitation at 535 nm) to PBS solution (ratio of 1:1:1000, v/v). Then, the fluorescence images of the cells were taken by a Leica DMI6000 B inverted microscope.

Animal experiments: All animal experiments were conducted under a protocol approved by the Institutional Animal Care and Use Committee of Southern University of Science and Technology, China. Sex was not considered in study design. The healthy male Sprague-Dawley rats (6–8 weeks old, 200–250 g) were used to implant the devices. Briefly, anesthesia was induced and maintained with inhalational 2% isoflurane. Once anesthetized, rats were fixed in a supine or prone position, and the surgical area was shaved and cleaned using 70% ethanol. The regions of dorsal, leg and chest were sterilized with iodine and alcohol scrubs for three times and a skin incision of ~1.5 cm was made. The devices ($2.5 \times 2.5\,cm^2$) were implanted between the epithelial and deep muscle layer. At last, the incision was sutured. Outputs of the stretchable device were recorded by connecting the probes to electrometer (Keithley 6514) via the exposed wires of implanted device. For long-term implantation, rats were prepped as above and a skin incision of ~1.5 cm was made in the dorsal region. The incision was closed, and animals were allowed to recover and return to normal housing for up to 10 weeks and the voltage output was recorded every 2 weeks. All animals were euthanized at the end of the studies.

Biocompatibility and biosafety assessment: Histological analysis was used for safety assessment of devices in rats at the 2nd and 10th week of post implantation, respectively. For histological analysis, after animals were euthanized at the end of study, tissue around device and most vital organs (heart, liver, spleen, lung and kidney) were collected for histological analysis. The tissues were first fixed in 4 vol% paraformaldehyde solution, dehydrated, and then embedded in paraffin wax using standard histopathological protocols. The tissues were sectioned into slides of 5 μm thickness using the Leica RM2016 Cryostat (Leica, Germany) and utilized Hematoxylin and Eosin (H&E) staining for analysis. The images of stained samples were captured by a slide scanner (NanoZoomer S60 C13210-01, Japan). Blood and serum tests were performed at the 2nd week of post implantation. Blood samples were taken after the animals were sedated and the analyses were performed by a 5-part Auto Hematology Analyzer (DF52Vet, Dymind, China).

### Reporting summary

Further information on research design is available in the Nature Portfolio Reporting Summary linked to this article.

## Data availability

The numerical data that support the findings of this study is available in figshare with the identifier "https://doi.org/10.6084/m9.figshare.26161780".

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

## Acknowledgements

This work was supported by the National Natural Science Foundation of China (No. 52272299), Shenzhen Science and Technology Program (No. 20220815141329003 and KQTD20180411143514543), Guangdong Provincial Department of Education Innovation Team Program (No. 2021KCXTD012), Guangdong-Hong Kong-Macau Joint Laboratory on Micro-Nano Manufacturing Technology (2021LSYS004). Y.S. was supported by the National Natural Science Foundation of China (No. 52303107) and Research Projects of Yancheng Institute of Technology (No. xjr2023029).

## Author contributions

Y.S. and Y.Y. conceived the idea and designed the experiments. Y.S. performed film fabrications, property measurements, and device assemblies and characterizations with help from Y.Z., C.L., R.X., X. X., H.Z., D.C., and H.W.; J.Y. did animal experiments with help from Y.S., L.L., G.C., and F.R.; Q.L., K.W., C.Y., and X.C. carried out finite element simulations. Y.S. and Y.Y. wrote the manuscript with contributions from all authors.

## Competing interests

The authors declare no competing interests.
