## [Peer Review File · Nature Communications]

A Universal Packaging Substrate for Mechanically Stable Assembly of Stretchable ElectronicsREVIEWER COMMENTS

Reviewer #1 (Remarks to the Author):

The authors developed a stretchable packaging substrate based on low water-permeable and chemically stable SIBS elastomers. The softness/stretchability of substrates was tuned over a wide range by changing PS content in SIBS and introducing PIB plasticizers. SIBS films with varied softness/stretchability could be combined into one substrate or packaging system by solvent welding process. The addition of PP-g-MAH in SIBS enabled its use as an interposer layer that can strongly adheres to both SIBS substrate through polymer chain entanglement and various materials with surfaces treated by aminosilane molecules through covalent bonding. The combination of module-specific, regional stretchability and tightly adhered interface prevented the delamination of modules with different modulus upon 600%. Moreover, the authors fabricated an arbitrarily deformable triboelectric generator/motion sensor based on the region-specific stretchability of combined SIBS substrates. Owing to the preserved low water permeability and chemical stability as well as the proven biocompatibility, the deformable triboelectric sensor showed stable performance over 10 weeks in vivo test.

SIBS has been known as one of the best packing materials among elastomers for stretchable electronics (<https://doi.org/10.1002/aenm.202103148>). Still, it is interesting to see that the low water permeability of SIBS remains the same even after introducing soft plasticizer and solvent welding. The concept of regional stretchability to incorporate both soft and hard modules into stretchable substrates was already demonstrated in ref. 17-22 as the authors have discussed in the introduction section. The difference in this work that I recognized is that the authors rendered mother elastomer (SIBS) soft (Elastic modulus < 0.5 MPa) yet still low water-permeable whereas others in ref. 17-22 reinforced mother elastomers (PDMS and PU) to create rigid island regions. Such material development as well as the universal interpose layer would enable a newly designed, fully deformable triboelectric motion sensor with excellent biocompatibility and stability. The manuscript is well concisely written, and the conclusions are supported by various experimental/simulation data. Considering the novelty and universal applicability of the developed substrate for stretchable electronics, I recommend the publication of this manuscript in Nature Communications after minor revisions according to the comments below.

- Could you please discuss in the main text why SIBS possesses remarkable impermeability compared with other elastomers?
- I don't agree with the statement below since other cases with regional stretchability also could accommodate the dynamic module in rather soft region.
"This strategy is effective in minimizing the mechanical influence of substrate deformation on the operation of the stationary module, but cannot apply to the dynamic module, which relies on the deformation to functionate."
- Could you please discuss the rationale behind the graph in Figure 1b? For example, reason behind why crack driving force (G) is higher for soft module than for hard module? Why G is higher with interposer layer?
- The Mises stress values in the main text (12.46 MPa for case 3, 4.65 MPa for case 4) are different to from what is shown in Figure 1d.
- (line 132), I am afraid if 50% is typo, and instead, 40% is correct.
- I am afraid if 0% in Figure 4d,e is typo, and instead, 10% is correct.

Reviewer #2 (Remarks to the Author):

The paper's concept is intriguing and sheds light on significant advancements in stretchable electronics. However, it appears challenging to identify originality concerning materials and processes. I suggest considering submission to more specialized journals.

Major Comments:

1. Accentuation of Distinctiveness and Novelty:

- Emphasize uniqueness and novelty compared to existing literature, particularly in lines 49-51 and 52-54. Additional references are needed to support these assertions.
- Specify differences from the previously presented gradient modulus substrate (ref. 1-3).
- Clearly define what factors the bioelectronic sensor can sense and how it differs from other sensors. Some graphs or tables could be utilized for clarity.

2. Elucidation of Theoretical Details:

- Provide more detailed explanations with accurate references, especially regarding the energy release rate (G) mentioned in lines 81-88.

3. Critical Evaluation of Main Data:

- Please scrutinize references cited in lines 122-125 (27, 28), ensuring their relevance to the chain mobility of plasticized elastomer.
- Validate the increase in polystyrene content and its effect on chain mobility. Generally, polystyrene has a much higher glass transition temperature (T_g) than polyisobutylene.
- It seems necessary to verify whether the absence of phase separation in the AFM data of the existing SIBS is not a measurement error.
- Additional analyses like differential scanning calorimetry (DSC) is recommended to measure changes in T_g .

4. Refinement of Methodological Framework:

- Please enhance the solvent welding process to ensure it produces clean surfaces and high resolution.

5. Demonstration of Mechanical Integrity:

- Include a demonstration illustrating the mechanical integrity of an actual operating circuit when stretched. The triboelectric generator and bioelectronic device do not support the main theme of this paper.

Minor Comments:

6. Illustration Improvement (Line 185):

- Magnify the strain 100% region in Figure 2e to provide a clearer depiction. Additionally, expand the elastic region in Figure S3 to illustrate modulus differences more effectively.

7. Enhancing Clarity (Line 214):

- Rectify the typographical error in line 214 and add an enlarged graph inset for Figure 4h to demonstrate cyclic operation more comprehensively.

8. Language Precision (Lines 244-245):

- Refrain from using terms like "presumably" and "speculate." Instead, focus on providing empirical evidence and theoretical support.

Reference:

1. N. Naserifar, et al. Material Gradients in Stretchable Substrates toward Integrated Electronic Functionality. *Adv. Mater.* 28, 18, 3584-3591. (2016)
2. R. Moser, et al. From Playroom to Lab: Tough Stretchable Electronics Analyzed with a Tabletop Tensile Tester Made from Toy-Bricks. *Adv. Sci.* 3, 4, 1500386. (2016)
3. R. Libanori, et al. Stretchable heterogeneous composites with extreme mechanical gradients. *Nat. Commun.* 3, 1265. (2012)

Response to comments

Reviewer 1:

The authors developed a stretchable packaging substrate based on low water-permeable and chemically stable SIBS elastomers. The softness/stretchability of substrates was tuned over a wide range by changing PS content in SIBS and introducing PIB plasticizers. SIBS films with varied softness/stretchability could be combined into one substrate or packaging system by solvent welding process. The addition of PP-g-MAH in SIBS enabled its use as an interposer layer that can strongly adheres to both SIBS substrate through polymer chain entanglement and various materials with surfaces treated by aminosilane molecules through covalent bonding. The combination of module-specific, regional stretchability and tightly adhered interface prevented the delamination of modules with different modulus upon 600%. Moreover, the authors fabricated an arbitrarily deformable triboelectric generator/motion sensor based on the region-specific stretchability of combined SIBS substrates. Owing to the preserved low water permeability and chemical stability as well as the proven biocompatibility, the deformable triboelectric sensor showed stable performance over 10 weeks in vivo test.

SIBS has been known as one of the best packing materials among elastomers for stretchable electronics (<https://doi.org/10.1002/aenm.202103148>). Still, it is interesting to see that the low water permeability of SIBS remains the same even after introducing soft plasticizer and solvent welding. The concept of regional stretchability to incorporate both soft and hard modules into stretchable substrates was already demonstrated in ref. 17-22 as the authors have discussed in the introduction section. The difference in this work that I recognized is that the authors rendered mother elastomer (SIBS) soft (Elastic modulus < 0.5 MPa) yet still low water-permeable whereas others in ref. 17-22 reinforced mother elastomers (PDMS and PU) to create rigid island regions. Such material development as well as the universal interpose layer would enable a newly designed, fully deformable triboelectric motion sensor with excellent biocompatibility and stability. The manuscript is well concisely written, and the conclusions are supported by various experimental/simulation data. Considering the novelty and universal applicability of the developed substrate for stretchable electronics, I recommend the publication of this manuscript in Nature Communications after minor revisions according to the comments below.

A: We thank the reviewer for the supportive recommendation. The reviewer's comments are addressed below point by point.

- Could you please discuss in the main text why SIBS possesses remarkable impermeability compared with other elastomers?

A: We thank the reviewer for the constructive suggestion. The remarkable impermeability of SIBS originated from its molecular composition and structure. SIBS is a triblock copolymer composed of a polyisobutylene (PIB) elastic block and polystyrene (PS) end-blocks. The fully saturated

nonpolar C-C backbone provides strong chemical repulsion to the polar water molecule. The dense methyl side groups in the PIB segments create a large steric hindrance, further preventing the penetration of water. In comparison, the Si-O bond in silicones has higher tendency to attract water molecules due to its polarity, leading to two-orders-of-magnitude higher water permeability relative to SIBS (*Prog. Polym. Sci.* 2001, 26, 985-1017). The hindrance effect of ethylene-butadiene segment in SEBS is weaker than that of the PIB segment, resulting in one-order-of-magnitude higher water permeability relative to SIBS (*Robotics* 2019, 8, 60).

To reveal this information, a new paragraph was added on page 7 of the main text as “The WVTR of SIBS is orders-of-magnitude lower than that of commonly used elastomers, such as SEBS ($\sim 5 \text{ g}\cdot\text{mm}\cdot\text{m}^{-2}\cdot\text{day}^{-1}$) and PDMS ($\sim 70 \text{ g}\cdot\text{mm}\cdot\text{m}^{-2}\cdot\text{day}^{-1}$). The low WVTR of SIBS originated from its molecular composition and structure. SIBS is a triblock copolymer composed of polystyrene (PS) end-blocks and a polyisobutylene (PIB) elastic block. The fully saturated nonpolar C-C backbone provides strong chemical repulsion to the polar water molecule. The dense methyl side groups in the PIB segments create a large steric hindrance, further preventing the penetration of water. Compared to SIBS, the Si-O bond in silicones has higher tendency to attract water molecules due to its polarity³⁴ and the steric hindrance effect of the ethylene-butadiene segment in SEBS is weaker than that of the PIB segment³⁵. The low WVTR of SIBS suggests they can more effectively protect devices from moisture-induced damages relative to SEBS and PDMS.”

- I don't agree with the statement below since other cases with regional stretchability also could accommodate the dynamic module in rather soft region.

“This strategy is effective in minimizing the mechanical influence of substrate deformation on the operation of the stationary module, but cannot apply to the dynamic module, which relies on the deformation to functionate.”

A: We thank the reviewer for the enlightening comment. This statement was indeed inaccurate if considering the integration of stretchable material networks (e.g. Ag nanowires) at the soft region (*Small* 2017, 13, 1700070; *Adv. Mater.* 2018, 30, 1704229). Our original intention was to compare our work with the category of packaged functional modules rather than the material networks, but the previous description was not sufficiently clear.

We agree with the reviewer's opinion on the difference between our work and previous works: “The difference in this work that I recognized is that the authors rendered mother elastomer (SIBS) soft (Elastic modulus < 0.5 MPa) yet still low water-permeable whereas others in ref. 15-22 reinforced mother elastomers (PDMS and PU) to create rigid island regions.” Inspired by this insightful analysis, we changed the statement mentioned by the reviewer on page 3 of the main text to “The dominant strategy for making area-dependent stretchability is to create rigid island regions in mother elastomers by introducing crosslinking or hard units, while softening mother elastomers without substantially altering their barrier properties remains a challenge²³⁻²⁵.”

- Could you please discuss the rationale behind the graph in Figure 1b? For example, reason behind why crack driving force (G) is higher for soft module than for hard module? Why G is higher with interposer layer?

A: We thank the reviewer for the detailed suggestion. Fig. 1b aims to provide a mechanical analysis for our substrate design. From the perspective of fracture mechanics, interface delamination is driven by the energy release rate (G), which relates to the strain and modulus of the module and substrate through the following equation: $G=Zh_m\sigma^2/E_m$, where σ is the tensile stress and $\sigma = E_s\varepsilon$, ε is the strain, h_m is the thickness of the module, E_m and E_s are the Young's modulus of the module and substrate, respectively, Z is a dimensionless driving force that determined by the cracking pattern and the modulus mismatch between the module and substrate (*Adv. Appl. Mech.* 1991, 29, 63-191). This equation indicates that G increases monotonically with the strain and the modulus mismatch between the module and substrate. Delamination occurs when G exceeds the interfacial toughness Γ (i.e., $G > \Gamma$, *Soft. Matter.* 2016, 12, 1093-1099). Therefore, improving the mechanical stability of the assembled modules on the substrate requires a low modulus mismatch for reducing G and a high interfacial toughness for increasing Γ .

For the reason behind why crack driving force (G) is higher for soft module than for hard module: a lower E_m yields a higher G according to the above equation. The modulus of the soft module is lower than that of the hard module, thus its G is higher.

For why G is higher with interposer layer: we need to clarify that the interposer layer does not affect G , but it increased Γ . The interposer layer enables stronger interfacial adhesion (i.e., higher interfacial toughness Γ), so that the interface can tolerate a larger strain and a higher G . G is an experimental parameter and Γ is a material property. They share the same dimension of energy per unit area.

To make these points clear, we rewrote the first paragraph in the section of Design Principle on page 4 of the main text as "From the perspective of fracture mechanics, interface delamination is driven by the energy release rate (G), which relates to the strain and modulus of the module and substrate through the following equation: $G=Zh_m\sigma^2/E_m$, where σ is the tensile stress and $\sigma = E_s\varepsilon$, ε is the strain, h_m is the thickness of the module, E_m and E_s are the Young's modulus of the module and substrate, respectively, Z is a dimensionless driving force that determined by the cracking pattern and the modulus mismatch between the module and substrate²⁶. This equation indicates that G increases monotonically with the strain and the modulus mismatch between the module and substrate. Delamination occurs when G exceeds the interfacial toughness Γ (i.e., $G > \Gamma$)²⁷. Therefore, improving the mechanical stability of the assembled modules on the substrate requires a low modulus mismatch for reducing G and a high interfacial toughness for increasing Γ ." The following sentences were added in the discussion of Figure 1b on page 5 of the main text as "For the case 4 in Fig. 1b, the lower E_m of the soft module leads to a higher G value relative to the hard module under the same strain according to the above equation. For the case 3 and 4 in Fig. 1b, the interposer

layer does not affect G , but it increased Γ . The interposer layer enables stronger interfacial adhesion (i.e., higher interfacial toughness Γ), so that the interface can tolerate a larger strain and a higher G .”

- The Mises stress values in the main text (12.46 MPa for case 3, 4.65 MPa for case 4) are different to from what is shown in Figure 1d.

A: We thank the reviewer for point this out. This discrepancy was a result of our inappropriate usage of the illustration scale in Fig. 1d. The maximum Mises stress calculated for the substrates in case 3 and case 4 concentrate at a small area (points B and B' in Fig. R1a, $\sigma_B = 12.46$ MPa and $\sigma_{B'} = 4.65$ MPa). It is difficult to tell the difference between these two cases under non-unified scales. To make them visible, we previously set the scale range to 5 MPa for both cases (Fig. R1b). After reading the reviewer’s comment, we realized a smarter choice would be unifying the scale range to 12.46 MPa for both cases with an enlarged figure illustration (Fig. R1c,d).

Based on the above consideration, Fig. R1d was used as the new Fig. 1d. One sentence was added to the caption of Fig. 1d to reveal the motivation of scale illustration as “The scale range for these two cases was unified to 12.46 MPa for a clear comparison.” Fig. R1c was added to Fig. S1 in supporting information. One sentence was added to the caption of Fig. 1d to reveal the motivation of scale illustration as “The scale ranges for the stress and strain fields were unified to 12.46 MPa and 8.69 for a clear comparison, respectively.”

Fig. R1 (new Fig. S1 and Fig. 1d). The stress fields for case 3 and 4 in Fig. 1d with (a) a non-unified scale, (b) a unified scale to 5 MPa, and (c) a unified scale to 12.46 MPa. (d) Enlarged

illustrations for the stress fields of case 3 and 4 at the soft/hard interface with a unified scale to 12.46 MPa.

- (line 132), I am afraid if 50% is typo, and instead, 40% is correct.

A: We thank the reviewer for pointing this out. The 50% was indeed a typo. It was changed to 40%.

- I am afraid if 0% in Figure 4d,e is typo, and instead, 10% is correct.

A: We thank the reviewer for pointing this out. They were indeed typos and were changed to 10% (new Fig. 4d, e).

Reviewer 2:

The paper's concept is intriguing and sheds light on significant advancements in stretchable electronics. However, it appears challenging to identify originality concerning materials and processes. I suggest considering submission to more specialized journals.

A: We thank the reviewer for appreciating the concept of our work. The discussions about the originality of this work on materials and processes were significantly improved after addressing the reviewer's comments.

Major Comments:

1. Accentuation of Distinctiveness and Novelty:

- Emphasize uniqueness and novelty compared to existing literature, particularly in lines 49-51 and 52-54. Additional references are needed to support these assertions.

A: We thank the reviewer for the valuable suggestion. Lines 49-51 refer to the existing adhesion designs for a packaging substrate. The previous attempts of module-substrate adhesions emphasize chemically alike interfaces, such as silicones/glass (*J. Adhes. Sci. Technol.* 2014, 28, 1046-1054; *Biosensors* 2021, 11, 292) and carbon elastomers/carbon elastomers (*Sci. Adv.* 2021, 7, eabg9180; *Nature* 2023, 614, 456-462). Adhesions between a module and a packaging substrate with significantly varied surface chemistry are seldomly explored (e.g., silicones vs. carbon elastomers,

Nat. Mater. 2016, 15, 190-196). One substrate that can universally adhere to different modules and intimately deform with them remains unavailable (*Natl. Sci. Rev.* 2023, 10, nwac172). The uniqueness and novelty of our work compared to lines 49-51 is to report a packaging substrate that adheres to a number of commonly used materials in stretchable electronics regardless of their surface chemistries, including PDMS, Al foil, Si, PEDOT: PSS, PET, and chitosan hydrogel (Table R2).

Lines 52-54 refer to the existing strain engineering for a packaging substrate. For this aspect, the Reviewer 1 provided an enlightening perspective that identified the difference between our work and previous reports – “The difference in this work that I recognized is that the authors rendered mother elastomer (SIBS) soft (Elastic modulus < 0.5 MPa) yet still low water-permeable whereas others in ref. 17-22 reinforced mother elastomers (PDMS and PU) to create rigid island regions.” We agree with this statement. The existing strategy for making area-dependent stretchability is indeed to create rigid island regions in mother elastomers by introducing crosslinking or hard units, while softening mother elastomers without substantially altering their barrier properties remains a challenge (*ACS Appl. Mater. Interfaces* 2022, 14, 18935–18943; *Chem. Eng. J.* 2023, 463,142350; *Adv. Energy Mater.* 2020, 10,2001424). The uniqueness and novelty of our work compared to lines 52-54 is to obtain a regional stretchability by softening the mother material with a minimal influence on the permeability, rather than introducing hard domains (Table R1).

Another uniqueness of our work is to synergistically design the interfacial adhesion and regional stretchability. These two aspects of packaging substrates were separately considered in previous literatures (*Adv. Mater.* 2021, 33, 2003155). Combining these two strategies allows us to achieve a stable module/substrate interface that can endure a record strain of 600% (Table R1). Our work is also the first demonstration of strain engineering on SIBS (Table R1).

To make these points clear, Table R2 was added in the supporting information as Table S2 and corresponding discussions were added on page 8 of the main text as “This is the first stretchable packaging substrate that can universally adhere metals, ceramics, plastics, and elastomers with distinct surface chemistries to the best of our knowledge (Table S2).” Table R1 was added in the supporting information as Table S1 and related discussions on page 3 of the main text was revised to “The dominant strategy for making area-dependent stretchability is to create rigid island regions in mother elastomers by introducing crosslinking or hard units, while softening mother elastomers without substantially altering their barrier properties remains a challenge²³⁻²⁵.” The discussion on page 5 of the main text was revised to “For the module-specific substrate with the interposer layer (i.e., a combination of bulk modulus and surface adhesion designs), both stiff and stretchable modules maintained their structural stability under 600% strain (case 4 in Fig. 1b and 1c), the largest strain degree for a module/substrate interface to the best of our knowledge (Table S1, Supplementary Information).” Following the reviewer’s suggestion, the following references (*ACS Appl. Mater. Interfaces* 2022, 14, 18935–18943; *Chem. Eng. J.* 2023, 463,142350; *Adv. Energy Mater.* 2020, 10, 2001424) were added to page 3 of the main text to support the above assertions.

Table R1. Comparison of packaging substrates with regional stretchability

Substrate	Strain (%)	Strategy	Modulus	Ref.
PDMS	300	Create rigid region by secondary cross-linking	0.17~0.6 MPa	1
PDMS	100	Create rigid region by UV exposure	0.12~2.76 MPa	2
PDMS	175	Create rigid region with transferring	0.12 MPa; 1.65 GPa	3
Ecoflex	70	Connect rigid and soft regions by welding	0.12~0.65 MPa	4
Ecoflex	150	Create rigid region by welding PDMS	50 kPa, 1.9 MPa	5
PU	10	Create rigid region by penetrating platelets	230 MPa; 1.3 GPa	6
PU	300	Create rigid region by penetrating platelets	4 MPa; 7GPa	7
PDMS	100	Create rigid region by embedding	0.2MPa;170GPa	8
SEBS	100	Connect rigid and soft regions by welding	1.5 MPa; 5 MPa	9
SEBS	500	Connect rigid and soft regions by welding	/	10
Polysulfone	24	Create rigid region by surface oxidating	5.3 MPa, 775 MPa	11
PDMS	70	Create rigid region with larger crosslink density	3 MPa; 160 MPa	12
PAAm-alginate	150	Create rigid region by locally stiffened hydrogel	40 kPa, 330 kPa	13
PDMS	40	Create rigid region by embedding	1 MPa; 1.5 GPa	14
PDMS	60	Create rigid region by embedding	1 MPa; 650 MPa	15
Ecoflex	100	Create rigid region by embedding	70 kPa, 2.7 GPa	16
PUA	60	Create rigid region by UV irradiation	/	17
PDMS	30	Create rigid region by partially perforated	0.5 MPa, 1GPa	18
Micro-holes	38	Introduce micro-holes	1 MPa	19
PDMS	20	Introduce protruding structure	1 MPa	20
PVDF-HFP	50	Create rigid region with microelectrode	93 kPa	21
SIBS	600	Create soft region with oligomer	60 kPa~3 MPa	This work

Abbreviations: polydimethylsiloxane (PDMS); Polyurethane (PU); Polystyrene–ethylene/butylene styrene)(SEBS); Polyacrylamide (PAAm); Polyurethane acrylate (PUA); Poly(vinylidene fluoride-co- hexafluoropropylene) (PVDF-HFP); Polystyrene-isobutylene-styrene (SIBS).

Table R2. Comparison of interfacial adhesions between module and stretchable substrate

Stretchable substrate	Materials		Strain (%)	Strategy	Ref.
	Adhered film				
Ecoflex	Hydrogel		125	Chemical bonding	22
Ecoflex	Hydrogel		700	Chemical bonding	23
SEBS	SEBS		100	Topological entanglements	24
SEBS	SEBS		30	Topological entanglements	25
PDMS	Glass		\	Chemical bonding	26
Au/PDMS	Au/PDMS		35	Mechanical interlocking	27
SIS	Rigid unit		500	Mechanical interlocking	28
PDMS	PDMS		80	Chemical bonding	29
PDMS	PET		33	Chemical bonding	30
PDMS	Organogel/Hydrogel		250	Mechanical interlocking	31
SBS	SIS		100	Topological entanglements	32
SEBS	SEBS		100	Topological entanglements	33
SEBS	SEBS		20	Topological entanglements	34
PDMS	PDMS/Ag flakes		80	Topological entanglements	35
PDMS	PEDOT:PSS		100	Hydrogen bonding	36
PU	PEDOT:PSS		80	Molecular interdiffusion	37
PDMS	Butyl rubber		80	Introducing micro-holes	38
SBS	PDMS		100	Physical effect	39
Ecoflex	Ecoflex		50	Molecular interdiffusion	40
PDMS based film	PDMS/SEBS/PU		60	Covalently and non-covalently bonding	41
SIBS	Si/PET/PDMS/Al foil/ PEDOT: PSS/ hydrogel		600	Topological entanglements and chemical bonding	This work

Abbreviations: Polydimethylsiloxane (PDMS) ; Poly(styrene-isoprene-styrene) (SIS); Poly(styrene-butadiene-styrene) (SBS); Poly(styrene-ethylene/butylene-styrene) (SEBS); Polyurethane (PU); Poly(styrene-isobutylene-styrene) (SIBS);

• Specify differences from the previously presented gradient modulus substrate (ref. 1-3).

1. N. Naserifar, et al. Material Gradients in Stretchable Substrates toward Integrated Electronic Functionality. *Adv. Mater.* 28, 18, 3584-3591. (2016)

2. R. Moser, et al. From Playroom to Lab: Tough Stretchable Electronics Analyzed with a Tabletop Tensile Tester Made from Toy-Bricks. *Adv. Sci.* 3, 4, 1500386. (2016)
3. R. Libanori, et al. Stretchable heterogeneous composites with extreme mechanical gradients. *Nat. Commun.* 3, 1265. (2012)

A: We thank the reviewer for the detailed suggestion. This comment is related to the previous comment about identifying the uniqueness of our work with references in Lines 52-54. The ref. 3 listed in this comment was originally cited in Lines 52-54. The strain engineering demonstrated in these three references was achieved by introducing hard domains to mother materials (i.e., Si islands to PDMS in ref. 1, rigid PDMS to Ecoflex in ref. 2, and Al₂O₃ microplatelets to PU in ref. 3), analogous to other references summarized in Table R1. As discussed in the previous response, the uniqueness and novelty of our work compared to lines 52-54 is to obtain a regional stretchability by softening the mother material with a minimal influence on the permeability, rather than introducing hard domains. Combining the design of advanced interfacial adhesion with the strain engineering is another difference of our work with these three references.

To further clarify this point, reference 1 and 2 were added to the introduction. References 1-3 were all included in Table R1. Related discussions on page 3 of the main text were revised to “The dominant strategy for making area-dependent stretchability is to create rigid island regions in mother elastomers by introducing crosslinking or hard units, while softening mother elastomers without substantially altering their barrier properties remains a challenge²³⁻²⁵.” The discussion on page 5 of the main text was revised to “For the module-specific substrate with the interposer layer (i.e., a combination of bulk modulus and surface adhesion designs), both stiff and stretchable modules maintained their structural stability under 600% strain (case 4 in Fig. 1b and 1c), the largest strain degree for a module/substrate interface to the best of our knowledge (Table S1, Supplementary Information).”

- Clearly define what factors the bioelectronic sensor can sense and how it differs from other sensors. Some graphs or tables could be utilized for clarity.

A: We thank the reviewer for the insightful suggestion. The bioelectronic sensor in our work can sense subtle biomechanical deformations, such as chest movements during respiration and thigh movements during leg bending. This sensor has two differences with reported implantable sensors. First, conventional biosensors can only work through a particular deformation mode (e.g., pressing or stretching), while this sensor can work with multiple deformation modes (e.g., pressing, stretching, bending, and crumbling) owing to its fully stretchable nature. Second, conventional biosensors have relatively short in vivo lifetime due to the ingress of biofluids, while this sensor achieves the longest in vivo lifetime (10 weeks) among stretchable implantable biosensors to the best of our knowledge. This longevity resulted from the low permeability of our newly developed packaging substrate (Fig. 2f).

Following the reviewer’s suggestion, we highlighted the difference of our work with literatures by adding a Table S3 in the supporting information (Table R3). A sentence for describing this

comparison was added on page 11 of the main text as “making the longest in vivo lifetime among the stretchable implantable biomechanical sensor to the best of our knowledge (Table S3, Supplementary Information).”

Table R3. Comparison of flexible/stretchable implantable biosensors

Encapsulation materials	In-vivo durability	Deformation response	Function	Ref.
PDMS-parylene	48 h	Compress	Cardiac monitoring	42
PLA	5 days	Compress	Pressure sensor	43
PLGA	14 days	Compress	Stimulation of nerve cells	44
Kapton	48 h	Compress	Cardiac monitoring	45
PDMS	5 days	Compress	\	46
Polyimide/Epoxy	48 h	Compress	Motility sensing	47
PLA	2 weeks	Compress	Monitoring the nerve tissue repair	48
PDMS-parylene	30 days	Compress	Cardiac monitoring	49
PLA	4 days	Compress	Respiration monitoring	50
PDMS-parylene	72 h	Stretch	Biomedical monitoring	51
PDMS-parylene	4 weeks	Compress	Cardiac monitoring	52
PDMS	3 days	Compress	modifying fibroblast alignment	53
PDMS-parylene	5 days	Compress	Heart motion	54
PLA	3 weeks	Compress	Force sensor	55
PDMS	3 days	Compress	Heart motion	56
Silicon rubber	24 h	Compress	Cardiac pacemaker	57
Ecoflex	24 h	Stretch	Respiration monitoring	58
Parylene-C	3 days	Compress	Blood glucose control	59
Silicon	7 days	Stretching	Motion monitoring	60
PDMS	7 days	Compress	\	61
SIBS	10 weeks	Stretch/bend compress/twist	Respiration monitoring	This work

Abbreviations: polydimethylsiloxane (PDMS); Poly(lactic acid) (PLA) ; Poly(lactic-co-glycolic acid) (PLGA); Poly(styrene-isobutylene-styrene) (SIBS).

2. Elucidation of Theoretical Details:

- Provide more detailed explanations with accurate references, especially regarding the energy release rate (G) mentioned in lines 81-88.

A: We thank the reviewer for the constructive suggestion. Following this suggestion, we enriched the discussion about energy release rate on page 4 of the main text as “From the perspective of fracture mechanics, interface delamination is driven by the energy release rate (G), which relates to the strain and modulus of the module and substrate through the following equation: $G=Zh_m\sigma^2/E_m$, where σ is the tensile stress and $\sigma = E_s\varepsilon$, ε is the strain, h_m is the thickness of the module, E_m and E_s are the Young’s modulus of the module and substrate, respectively, Z is a dimensionless driving force that determined by the cracking pattern and the modulus mismatch between the module and substrate²⁶. This equation indicates that G increases monotonically with the strain and the modulus mismatch between the module and substrate. Delamination occurs when G exceeds the interfacial toughness Γ (i.e., $G > \Gamma$)²⁷. Therefore, improving the mechanical stability of the assembled modules on the substrate requires a low modulus mismatch for reducing G and a high interfacial toughness for increasing Γ .” The following references about the energy release rate were added as new ref. 26, 27 in the main text: *Adv. Appl. Mech.* 1991, 29, 63-191; *Soft Matter*, 2015, 11, 882.

3. Critical Evaluation of Main Data:

- Please scrutinize references cited in lines 122-125 (27, 28), ensuring their relevance to the chain mobility of plasticized elastomer.

A: We thank the reviewer for pointing this out. The references in lines 122-125 (27, 28) were cited to indicate the existence of the surface mobile layer in polymers. They are indeed not closely relevant to the chain mobility of plasticized elastomer. Following this suggestion, we replaced them with references *Compos. Sci. Technol.* 2020, 195, 108202 and *Polymer* 2014, 55, 3905-3914. As shown below, the new references were directly linked with the chain mobility of plasticized elastomer.

New ref. 29: Ni. Y. et al. Plasticizer-induced enhanced electromechanical performance of natural rubber dielectric elastomer composites. *Compos. Sci. Technol.* 195,108202 (2020).

New ref. 30: Bhadane, P.A., Tsou, A.H., Cheng, J., Ellul, M.D., Favis, B.D. Enhancement in interfacial reactive compatibilization by chain mobility. *Polymer* 55, 3905-3914 (2014).

- Validate the increase in polystyrene content and its effect on chain mobility. Generally, polystyrene has a much higher glass transition temperature (T_g) than polyisobutylene.

A: We thank the reviewer for the insightful comment. Following this suggestion, we performed the DMA characterization for a pristine SIBS with 30 wt% PS. The testing parameters (e.g., temperature range, heating rate, strain amplitude, and strain frequency) were set to be identical with the measurements of the pristine SIBS with 15 wt% PS and the SIBS/PIB blend film (Fig. 2b).

As shown in Fig. R2, the pristine SIBS with 30 wt% PS exhibits two major transitions, including a temperature transition at $-28\text{ }^{\circ}\text{C}$ associated with the PIB segments and a temperature transition at $102\text{ }^{\circ}\text{C}$ associated with the PS segments. Compared with the pristine SIBS with 15 wt% PS and SIBS/PIB blend film, T_g of PIB segments in the SIBS with 30 wt% PS shifts to higher temperature due to the physical confinement induced by the PS hard domains. The T_g of the PS segments in the $\tan \delta$ was observed in the SIBS with 30 wt% PS, but not in the SIBS with 15 wt% PS and SIBS/PIB blends. Previous reports have shown that the T_g peak of the PS segment only appears in the $\tan \delta$ curve for PS-based block copolymers with $>30\text{ wt}\%$ PS (*J. Saudi Chem. Soc.* 2021, 25, 101211). For PS-based block copolymers with $<30\text{ wt}\%$ PS, sample softening and associated loss of its initial geometry is the main reason that T_g peak of the PS segment cannot be detected by DMA (*Polymer* 1997, 38, 4325-4335). However, the T_g of PS for the SIBS with 15 wt% PS and the SIBS/PIB blend film can be detected by differential scanning calorimetry (Fig. R4).

To clarify this point, Fig. R2 was added in the new Fig. S4 in supporting information. Corresponding discussions were added on page 6 of the main text as “The glass transition temperature (T_g) of SIBS was reduced from $-34\text{ }^{\circ}\text{C}$ and $-39\text{ }^{\circ}\text{C}$ after introducing 20 wt% of PIB, further evidencing the enhanced molecular mobility with the plasticizing effect of PIB oligomer^[29,30]. Due to the low content of PS, T_g of the PS segments was not observed on both samples in the $\tan \delta$ spectra, but appeared in the SIBS with 30 wt% PS (Fig. 2b and S4), consistent with previous observations on the PS-based block copolymers using dynamic mechanical analysis^[31,32].”

Fig. R2 (new Fig. S4). $\tan \delta$ spectra as a function of temperature for pristine SIBS with 30 wt% of PS, pristine SIBS with 15 wt% of PS, and SIBS/PIB blend.

- It seems necessary to verify whether the absence of phase separation in the AFM data of the existing SIBS is not a measurement error.

A: We thank the reviewer for the constructive comment. Following this suggestion, we performed a series of AFM characterizations to compare the surfaces of a pristine SIBS film and a SIBS/PIB blend film under varied AFM conditions. We found that the surface morphology of the pristine SIBS film depends on the measurement parameters. For example, when the driving amplitude of the AFM tip was low (150 mV), a vague picture was received on the pristine SIBS as shown in Fig. R3a (i.e., original Fig. 2a in the main text). When the driving amplitude of the AFM tip increased to 280 mV, a clear picture that identifies the PS domains was obtained on the pristine SIBS (Fig. R3b). In contrast, a clear phase image was obtained on the SIBS/PIB sample under both low and high driving amplitude of the AFM tip, implying a different surface property compared to the pristine SIBS (Fig. R3c,d). Previous reports have discovered that the surface feature of PS-based block copolymers under AFM depends on the equipment parameters that control the interactions (e.g., depth and contact area) between AFM tip and the sample (*Macromolecules* 2001, 34, 4159-4165; *ACS Nano* 2013, 7, 10387-10396; *Sci. Adv.* 2023, 9, eadg8292). This is consistent with our observations.

According to the phase images of the pristine SIBS and SIBS/PIB blend film acquired under 280 mV of the driving amplitude (Fig. R3b,d), the distance between PS blocks in the SIBS/PIB blend film (42 nm, Fig. R3d,f) was higher than that of the pristine SIBS film (24 nm, Fig. R3b,e). This phenomenon resulted from the plasticizing effect of PIB oligomers, which led to enlarged space between PS domains by disentangling SIBS chains.

To make these points clear, Fig. R3b was used as the new upper left graph of Fig. 2a in the main text, and Fig. R3e,f were added in the supporting information as the new Fig. S3. The discussion about the AFM characterizations on page 6 in the main text was revised as: “The distance between the PS blocks of SIBS was enlarged from 24 nm to 42 nm after adding PIB, implying the disentanglement of SIBS chains due to the plasticizing effect induced by the PIB oligomer (Fig. 2a and S3).”

The influence of the driving amplitude of AFM tip on the surface feature of the SIBS samples was revealed in the structure and property characterizations of Experimental Section in the supporting information as “The driving amplitude of AFM tip was set to 280 mV for the phase image in Fig. 2a. Under a lower driving amplitude (150 mV), the phase image of the pristine SIBS became vague, while the phase image of the SIBS/PIB blend remained clear.”

Fig. R3 (new Fig. 2a and S3). AFM characterization of a pristine SIBS film and a SIBS/PIB blend film. (a,b) AFM phase images of a pristine SIBS film acquired under 150 mV (a) and 280 mV (b) of the driving amplitude of AFM tip. (c,d) AFM phase images of a SIBS/PIB blend film acquired under 150 mV (c) and 280 mV (d) of the driving amplitude of AFM tip. (e,f) Statistical results of PS block distance for pristine SIBS (e) and SIBS/PIB blend (f). The weight percent of PS is 15% in these samples.

- Additional analyses like differential scanning calorimetry (DSC) is recommended to measure changes in T_g .

A: We thank the reviewer for the constructive suggestion. Following this suggestion, we performed DSC measurements for characterizing T_g of the pristine SIBS with 15 wt% PS, 30 wt% PS, and the SIBS/PIB blend film using a DSC1 (Mettler Toledo). As shown in Fig. R4, a step change in heat capacity was observed on all three samples at temperatures ranging from 104 °C to 110 °C, corresponding to the glass transition of PS segments. The T_g of the PIB segment was not present due to the lowest temperature limitation of our equipment (-60 °C). To detect the T_g of the PIB segment, the testing temperature needs to start from a significantly lower value (e.g., -100 °C) compared to the T_g of the PIB segment to cover the full temperature range of the glass transition (-70 °C to 30 °C, Fig. R2) (Polymers 2022, 14, 3742).

Fig. R4 was added as the new Fig. S5 in the supporting information, the corresponding discussion was added in page 6 of the main text as: “Due to the low content of PS, T_g of the PS segments was not observed on both samples in the $\tan \delta$ spectra, but appeared in the SIBS with 30 wt% PS (Fig. 2b and S4), consistent with previous observations on the PS-based block copolymers using dynamic mechanical analysis ^[31,32]. However, the T_g of PS for all samples was detectable on differential scanning calorimetry (Fig. S5). Experimental details about the DSC characterization were added in the section of “Differential scanning calorimetry” in the supporting information as: “DSC

measurements were performed on DSC1 (Mettler Toledo) with a lowest temperature of -60 °C. The samples were heated to 150 °C to remove heat history and quenched at 50 °C for 30 minutes. About 10 mg of the sample was used for the measurement. The sample was maintained at -60 °C for 10 minutes, and then heated from -60 °C to 120 °C with a heating rate of 10 °C/min under continuous nitrogen purge.

Fig. R4 (new Fig. S5). DSC curves of pristine SIBS with 30 wt% of PS, pristine SIBS with 15 wt% of PS, and SIBS/PIB blend. The rectangular highlights the region of the PS glass transition.

4. Refinement of Methodological Framework:

- Please enhance the solvent welding process to ensure it produces clean surfaces and high resolution.

A: We thank the reviewer for the constructive suggestion. As shown in Fig. R5a, the soft-hard film fabricated by the solvent welding method has a clean surface. Following the reviewer's suggestion, we improved the resolution by controlling the solvent evaporation during the solvent welding process. The physical size of the transition area between soft and hard regions (i.e., the resolution) was determined by the diffusion degree of the SIBS/PIB blend precursor (i.e., the soft region precursor) to the pre-solidified SIBS with 30 wt% PS (i.e., the hard region). This diffusion degree depends on the solidification degree of the SIBS in the hard region. The solidification degree can be further controlled by adjusting the time/degree of the solvent evaporation. Fig. R5 b-e shows the 3D confocal fluorescence images of soft-hard films that were fabricated with different evaporating times of toluene (hard region), including 3 h, 8 h, 12 h, and 24 h. The transition distance decreased with the increase of the solvent evaporation (i.e., the solidification degree of the SIBS). The smallest transition distance obtained on the 24 h sample was about 100 μm .

Fig. R5 was added as the new Fig. S7. Corresponding discussion was added on page 7 of the main text as “The surface of the transition area between soft and hard regions was clean (Fig. S7a). The physical size of the transition area (i.e., the resolution) depends on the solvent evaporating time of

the hard region, which controls the solidification degree of the hard region (Fig. S7b-f). The smallest transition distance between the soft and hard regions was about 100 μm .”

The experimental detail about the optimization of interfacial resolution was added in the section of “Fabrication of the packaging substrate with regional stretchability” and “Confocal laser scanning microscope” in the supporting information as “The optimization of the soft/hard interface shown in Fig. S7 was done by controlling the toluene evaporating time of the hard region. In specific, 4.5 mL of 250 mg/mL SIBS (30 wt% PS)/toluene solution was dropped in the left part of the rectangular container. A PDMS film was inserted in the right part to leave the space for the soft component. The solidification degree of the hard region in the left parts of the container were manipulated by varying the toluene evaporation at room temperature in the fume hood. After the toluene evaporated for different times ranging from 0 to 24 h, the PDMS spacer was removed and the blank space was filled with 4 mL SIBS/PIB/toluene solution. The hard-soft (HS) films with different transition interfaces were obtained after the evaporation of toluene.” and “Confocal fluorescence images were acquired on a Zeiss LSM 980 instrument. The top-view confocal images were obtained by a raster scan at 400 Hz on the x-y plane with a 5 \times air objective. The images were collected at a resolution of 1024 \times 1024 pixels and a depth of 16-bit. The SIBS/PIB/toluene solution was dyed with coumarin 6. The horizontal Z-stacked images of soft-hard were acquired by the reconstruction of cross-sectional images. All of the confocal images were analyzed using Zen Blue 3.0 software.”

Fig. R5 (new Fig. S7). Optimization of the interface between the soft and hard regions. (a) Top-view SEM image of the interface. (b-e) 3D confocal fluorescence images of the interfaces obtained at different evaporation times for the hard region: (b) 3 h, (c) 8 h, (d) 12 h, and (e) 24 h. The soft region was dyed in green and the hard region was transparent. (f) Transition distance between the soft and hard regions at a function of toluene evaporating time measured from the confocal fluorescence images in (b-e).

5. Demonstration of Mechanical Integrity:

- Include a demonstration illustrating the mechanical integrity of an actual operating circuit when stretched. The triboelectric generator and bioelectronic device do not support the main theme of this paper.

A: We thank the reviewer for the insightful comment. Following this suggestion, we demonstrated the steady operation of an actual circuit under large strains using our packaging substrate (Fig. R6 and Movie S1). The circuit includes stiff modules (e.g., LED, inductor, chip, diode, and resistor) and stretchable conducting lines made from a SIBS, PIB, and liquid metal composite. The stiff modules and stretchable conducting lines were integrated onto the hard and soft regions of the packaging substrate, respectively. When the circuit was externally powered and stretched, the LED remained to be lightening at the stretch of 400%. Further increasing the stretch made the LED go out due to the deformation-induced electrical breakdown of the stretchable conducting lines. However, the circuit remained to be structurally stable, consistent with the demonstration shown in Fig. 1c. The bioelectronic device shown in Fig. 4 and Fig. 5 demonstrated the overall packaging performance of our packaging substrate, including mechanical integrity, water prevention, and biocompatibility.

Fig.R6 was added as the new Fig. S2. Corresponding discussion was added on page 5 of the main text as an individual paragraph “The high mechanical integrity leads to the steady operation of an actual operating circuit under large strains (Fig. S2 and Movie S1). The circuit includes stretchable conducting lines made from a SIBS/PIB/liquid metal composite and stiff modules such as light emitting diode (LED), inductor, chip, diode, and resistor. The stiff modules and stretchable conducting lines were integrated onto the hard and soft regions of the packing substrate, respectively. When the circuit was externally powered and stretched, the LED remained lit up under a stretch of 400%. Further increasing the stretch made the LED go out due to the deformation-induced electrical breakdown of the stretchable conducting lines.”

The experimental detail about the circuit fabrication was added in the section of “Fabrication of the stretchable circuit” in the supporting information as “The stretchable conducting lines were fabricated using a composite of SIBS, PIB, and liquid metal. Initially, 1.35 g of liquid metal was introduced into 30 mL of acetone. This mixture underwent ultrasonication at 44% amplitude for 20 minutes in a water bath at room temperature to generate liquid metal microdroplets. Subsequently, the microdroplets were isolated via centrifugation at 2200 rpm for 20 minutes, and the solvent was carefully decanted. The resulting liquid metal microdroplets were then combined with 1 mL of SIBS/PIB/toluene solution (100 mg/mL) before being deposited onto a glass plate as ink. Upon complete evaporation of the toluene solvent, a composite film comprising SIBS, PIB, and liquid metal was obtained. This film was cut into strips and affixed to the soft region of the prepared SIBS substrate to serve as conducting lines. To increase their conductivity, an acoustic field was applied to the conducting lines using a probe sonicator at 30% amplitude. The stiff modules, including LED, inductor, IC chip, diode, and resistors, underwent treatment with oxygen plasma at 300 W for 5 minutes. Subsequently, their surfaces were treated with a solution of ethanol and water (9:1 volume

ratio) containing 5 wt% APTES, followed by drying at 80 °C for 1 hour. The treated modules were integrated onto module-specific SIBS substrates with an interposer layer.”

Fig. R6 (new Fig. S2). Digital images showing the mechanical and electrical stabilities of an actual operating circuit under large stretch.

Minor Comments:

6. Illustration Improvement (Line 185):

- Magnify the strain 100% region in Figure 2e to provide a clearer depiction. Additionally, expand the elastic region in Figure S3 to illustrate modulus differences more effectively.

A: We thank the reviewer for the detailed suggestions. The enlarged graphs of Fig. 2e and S3 was provided in the revised manuscript and supporting information.

Fig. R7a was used as the new Fig. 2e. The following description was added to the figure caption “The inset is a magnified illustration for the region within 100% of strain”. The discussion about Fig. 2e on page 6 of the main text was slightly revised to “The stress-strain curves showed a similar mechanical behaviour determined by the soft region for the soft film, SHS film, and HSH film when the strain was smaller than 100% (inset in Fig. 2e). With the increase of stress and strain, the hard regions experienced the load and differentiated the mechanical behaviours of these films (Fig. 2e)”.

Fig. R7b was used as the new Fig. S6. The following description was added to the figure caption “The inset is a magnified illustration for the region within 30% strain. The elastic modulus was determined from the slope within the linear range shown in the inset.”

Fig. R7 (new Fig. 2e and S6). Mechanical characterizations. (a) Stress-strain curves of hard, soft, HSH, and SHS films. The inset is a magnified illustration for the region within 100% of strain. (b) Stress-strain curve of the SIBS/PIB blend film, showing the decrease of modulus as the increase of PIB content. The inset is a magnified illustration for the region within 30% strain. The elastic modulus was determined from the slope within the linear range shown in the inset.

7. Enhancing Clarity (Line 214):

- Rectify the typographical error in line 214 and add an enlarged graph inset for Figure 4h to demonstrate cyclic operation more comprehensively.

A: We thank the reviewer for pointing these out. The typographical error in line 214 has been revised. Three enlarged graphs at the beginning, middle, and end of the testing period for Fig. 4h were included (Fig. R8). One sentence describing the output stability of the device was added on page 10 of the main text as “After 15,000 cycles of continuous deformation to 70%, the voltage output of the device remained at ~3 V (Fig. 4h)”. One sentence was added to the figure caption as “The inserts show the voltage output of the device at the beginning, middle, and end of the test.”

Fig. R8 (new Fig. 4h). Output stability of the device under 15, 000 cycles of stretching to 70%. The inserts show the voltage output of the device at the beginning, middle, and end of the test.

8. Language Precision (Lines 244-245):

- Refrain from using terms like "presumably" and "speculate." Instead, focus on providing empirical evidence and theoretical support.

A: We thank the reviewer for the detailed suggestion. Following this comment, the term “presumably” on page 7 of the main text was deleted. Two references were added to support the related discussion (*ACS Appl. Mater. Interfaces* 2022, 14, 18935-18943; *J. Member. Sci.* 2006, 280,427-432) as “The WVTR of the SIBS film with 60 wt% of PIB abruptly increased to 0.52 g·mm·m⁻²·day⁻¹, due to the new free volume created by the dominant PIB oligomers^[24,33].”

The sentence containing the term “speculate” on page 10 of the main text was revised to “Therefore, this packaging substrate and the device assembled from it will be biocompatible.” This statement is supported by the experimental results shown in Fig. 5.

References for Tables 1-3 (added in the supporting information):

1. Miao, L. et al. Localized modulus-controlled PDMS substrate for 2D and 3D stretchable electronics. *J. Micromech. Microeng.* **30**, 045001 (2020).
2. Cai, M., Nie, S., Du, Y., Wang, C., Song, J. Soft elastomers with programmable stiffness as strain-isolating substrates for stretchable electronics. *ACS Appl. Mater. Interfaces* **11**, 14340-14346 (2019).
3. Lee, Y. et al. Stretchable array of CdSe/ZnS quantum-dot light emitting diodes for visual display of bio-signals. *Chem. Eng. J.* **427**, 130858 (2022).
4. Yoon, J. et al. Design and fabrication of novel stretchable device arrays on a deformable polymer substrate with embedded liquid-metal interconnections. *Adv. Mater.* **26**, 6580-6586 (2014).
5. Moser, R. et al. From playroom to lab: tough stretchable electronics analyzed with a tabletop tensile tester made from toy-bricks. *Adv. Sci.* **3**, 1500386 (2016).
6. Erb, R. M. et al. Locally reinforced polymer-based composites for elastic electronics. *ACS Appl. Mater. Interfaces* **4**, 2860-2864 (2012)
7. Libanori, R. et al. Stretchable heterogeneous composites with extreme mechanical gradients. *Nat. Commun.* **3**, 1265 (2012).
8. Naserifar, N. et al. Material gradients in stretchable substrates toward integrated electronic functionality. *Adv. Mater.* **28**, 3584-3591 (2016).
9. Wang, W. et al. Strain-insensitive intrinsically stretchable transistors and circuits. *Nat. Electron.* **4**, 143-150 (2021).
10. Jiang, Y. et al. A universal interface for plug-and-play assembly of stretchable devices. *Nature* **614**, 456-462 (2023).
11. Cao, Y. et al. Direct fabrication of stretchable electronics on a polymer substrate with process - integrated programmable rigidity. *Adv. Funct. Mater.* **28**, 1804604 (2018).

12. Park, C. W. et al. Locally-tailored structure of an elastomeric substrate for stretchable circuits. *Semicond. Sci. Technol.* **31**, 025013 (2016)
13. Liu, H. et al. Spatially modulated stiffness on hydrogels for soft and stretchable integrated electronics. *Mater. Horiz.* **7**, 203-213 (2020)
14. Byun, J. et al. Fully printable, strain-engineered electronic wrap for customizable soft electronics. *Sci. Rep.* **7**, 45328 (2017).
15. Byun, J.; Chung, S.; Hong, Y. Artificial soft elastic media with periodic hard inclusions for tailoring strain-sensitive thin-film responses. *Adv. Mater.* **30**, 1802190 (2018).
16. Lim, Y. et al. Biaxially stretchable integrated array of high performance microsupercapacitors. *ACS Nano* **11**, 11639–11650 (2014)
17. Kim, Y.; Jun, S.; Ju, B. K.; Kim, J. W. Heterogeneous configuration of a Ag nanowire/polymer composite structure for selectively stretchable transparent electrodes. *ACS Appl. Mater. Interfaces* **9**, 7505-7514 (2017)
18. Lee, Y. K. et al. Chemical sensing systems that utilize soft electronics on thin elastomeric substrates with open cellular designs. *Adv. Funct. Mater.* **27**, 1605476 (2017)
19. Rong, Y. et al. Stretchability improvement of flexible electronics by laser micro-drilling array holes in PDMS film. *Opt. Lasers Eng.* **134**, 160307 (2020)
20. Cantarella, G. et al. Design of engineered elastomeric substrate for stretchable active devices and sensors. *Adv. Funct. Mater.* **28**, 1705132 (2018)
21. Su Q. et al. A stretchable and strain-unperturbed pressure sensor for motion interference-free tactile monitoring on skins. *Sci. Adv.* **7**, eabi4563 (2021)
22. Liu, T. et al. Triboelectric-nanogenerator-based soft energy-harvesting skin enabled by toughly bonded elastomer/hydrogel hybrids. *ACS Nano* **12**, 2818-2826 (2018).
23. Yuk, H., Zhang, T., Parada, G.A., Liu, X., Zhao, X. Skin-inspired hydrogel-elastomer hybrids with robust interfaces and functional microstructures. *Nat. Commun.* **7**, 12028 (2016).
24. Wang W. et al. Strain-insensitive intrinsically stretchable transistors and circuits. *Nat. Electron.* **4**, 143-150 (2021).
25. Jiang Y. et al. A universal interface for plug-and-play assembly of stretchable devices. *Nature* **614**, 456-462 (2023).
26. Xiong L., Chen P., Zhou Q. Adhesion promotion between PDMS and glass by oxygen plasma pre-treatment. *J. Adhes. Sci. Technol.* **28**, 1046-1054 (2014).
27. Zhu, M. et al. A mechanically interlocking strategy based on conductive microbridges for stretchable electronics. *Adv. Mater.* **34**, e2101339 (2022).
28. Lopes, P.A., Santos, B.C., de Almeida, A.T., Tavakoli, M. Reversible polymer-gel transition for ultra-stretchable chip-integrated circuits through self-soldering and self-coating and self-healing. *Nat. Commun.* **12**, 4666 (2021).
29. Hwang, H. et al. Stretchable anisotropic conductive film (S-ACF) for electrical interfacing in high-resolution stretchable circuits. *Sci. Adv.* **7**, eabh0171 (2021).
30. Erlenbach, S. et al. Flexible-to-stretchable mechanical and electrical interconnects. *ACS Appl. Mater. Interfaces* **15**, 6005-6012 (2023).
31. Jing, T. et al. Interfacial roughness enhanced gel/elastomer interfacial bonding enables robust and stretchable triboelectric nanogenerator for reliable energy harvesting. *Small* **19**, 2206528 (2023).

32. Song, W.J. et al. Stand - alone intrinsically stretchable electronic device platform powered by stretchable rechargeable battery. *Adv. Funct. Mater.* **30**, 2003608 (2020).
33. Wang, S. et al. Skin electronics from scalable fabrication of an intrinsically stretchable transistor array. *Nature* **555**, 83-88 (2018).
34. Molina-Lopez, F. et al. Inkjet-printed stretchable and low voltage synaptic transistor array. *Nat. Commun.* **10**, 2676 (2019).
35. Guo, W. et al. Matrix-independent highly conductive composites for electrodes and interconnects in stretchable electronics. *ACS Appl. Mater. Interfaces* **11**, 8567-8575 (2019).
36. Li, G. et al. PEDOT:PSS/grafted-PDMS electrodes for fully organic and intrinsically stretchable skin-like electronics. *ACS Appl. Mater. Interfaces* **11**, 10373-10379 (2019).
37. Lee, J. et al. Inter-diffused thermoplastic urethane-PEDOT:PSS bilayers with superior adhesion properties for high-performance and intrinsically-stretchable organic solar cells. *J. Mater. Chem. A* **11**, 12846-12855 (2023).
38. Vohra, A., Schlingman, K., Carmichael, R. S., Carmichael, T.B. Membrane-interface-elastomer structures for stretchable electronics. *Chem* **4**, 1673-1684 (2018).
39. You, I., Kong, M., Jeong, U. Block Copolymer Elastomers for Stretchable Electronics. *Acc. Chem. Res.* **52**, 63-72 (2019).
40. Jeong, K. et al. A sub-micron-thick stretchable adhesive layer for the lamination of arbitrary elastomeric substrates with enhanced adhesion stability. *Chem. Eng. J.* **429**, 132250 (2022).
41. Kang, J. et al. Tough-interface-enabled stretchable electronics using non-stretchable polymer semiconductors and conductors. *Nat. Nanotechnol.* **17**, 1265-1271 (2022).
42. Zheng, Q. et al. In vivo self-powered wireless cardiac monitoring via implantable triboelectric nanogenerator. *ACS Nano* **10**, 6510-6518 (2016).
43. Ouyang, H. et al. A bioresorbable dynamic pressure sensor for cardiovascular postoperative care. *Adv. Mater.* **33**, e2102302 (2021).
44. Zheng, Q. et al. Biodegradable triboelectric nanogenerator as a life-time designed implantable power source. *Sci. Adv.* **2**, e1501478 (2016).
45. Azimi, S. et al. Self-powered cardiac pacemaker by piezoelectric polymer nanogenerator implant. *Nano Energy* **83**, 105781 (2021).
46. Yu, Y. et al. Biocompatibility and in vivo operation of implantable mesoporous PVDF-based nanogenerators. *Nano Energy* **27**, 275-281 (2016).
47. Dagdeviren, C, et al. Flexible piezoelectric devices for gastrointestinal motility sensing. *Nat. Biomed. Eng.* **1**, 807-817 (2017).
48. Wu, P. et al. Ultrasound-driven in vivo electrical stimulation based on biodegradable piezoelectric nanogenerators for enhancing and monitoring the nerve tissue repair. *Nano Energy* **102**, 107707 (2022).
49. Li J, et al. Long-term in vivo operation of implanted cardiac nanogenerators in swine. *Nano Energy* **90**, 106507 (2021).
50. Yang, F. et al. Wafer-scale heterostructured piezoelectric bio-organic thin films. *Science* **373**, 337-342 (2021).
51. Ma, Y. et al. Self-powered, one-stop, and multifunctional implantable triboelectric active sensor for real-time biomedical monitoring. *Nano Lett.* **16**, 6042-6051 (2016).

52. Xie, F. et al, An experimental study on a piezoelectric vibration energy harvester for self-powered cardiac pacemakers. *Ann. Transl. Med.* **9**, 880 (2021).
53. Wang, A. et al. Piezoelectric nanofibrous scaffolds as in vivo energy harvesters for modifying fibroblast alignment and proliferation in wound healing. *Nano Energy* **43**, 63-71 (2018)
54. Li, N. et al. Direct powering a real cardiac pacemaker by natural energy of a heartbeat. *ACS Nano* **13**, 2822-2830 (2019)
55. Cheng, Y. et al. Boosting the piezoelectric sensitivity of amino acid crystals by mechanical annealing for the engineering of fully degradable force sensors. *Adv. Sci.* **10**, e2207269 (2023)
56. Ouyang, H. et al. Symbiotic cardiac pacemaker. *Nat Commun.* **10**, 1821 (2019)
57. Ryu, H. et al. Self-rechargeable cardiac pacemaker system with triboelectric nanogenerators. *Nat Commun.* **12**, 4374 (2021)
58. Li, J. et al. implanted battery-free direct-current micro-power supply from in vivo breath energy harvesting. *ACS Appl. Mater. Interfaces* **10**, 42030-42038 (2018)
59. Liu, Z. et al. A self-powered optogenetic system for implantable blood glucose control. *Research* **2022**, 9864734 (2022)
60. Sheng, F. et al. Ultrastretchable organogel/silicone fiber-helical sensors for self-powered implantable ligament strain monitoring. *ACS Nano* **16**, 10958-10967 (2022)
61. Cheng, B. et al. Mechanically asymmetrical triboelectric nanogenerator for self - powered monitoring of in vivo microscale weak movement. *Adv. Energy Mater.* **10**, 2000827 (2020)

REVIEWERS' COMMENTS

Reviewer #1 (Remarks to the Author):

The revised manuscript was significantly improved after addressing all comments from both reviewers.

Just one more question about the authors' sentence in the rebuttal letter, "we need to clarify that the interposer layer does not affect G , but it increased Γ . The interposer layer enables stronger interfacial adhesion (i.e., higher interfacial toughness Γ), so that the interface can tolerate a larger strain and a higher G ".

In such case, shouldn't the G slopes of case 1 and case 3 in Figure 1b be the same while Γ is different?

As long as this concern is addressed, I recommend the revised version for the publication in Nature Communications.

Reviewer #2 (Remarks to the Author):

Authors have addressed my questions and concerns satisfactory. I recommend to publish this work in Nature Communications.

Response to comments

Reviewer #1 (Remarks to the Author):

The revised manuscript was significantly improved after addressing all comments from both reviewers.

Just one more question about the authors' sentence in the rebuttal letter, "we need to clarify that the interposer layer does not affect G , but it increased Γ ". The interposer layer enables stronger interfacial adhesion (i.e., higher interfacial toughness Γ), so that the interface can tolerate a larger strain and a higher G ".

In such case, shouldn't the G slopes of case 1 and case 3 in Figure 1b be the same while Γ is different?

As long as this concern is addressed, I recommend the revised version for the publication in Nature Communications.

A: We thank the reviewer for the supportive recommendation and the critical suggestion. Our previous statement "the interposer layer does not affect G , but it increased Γ " was incorrect. A more accurate statement should be "the interposer layer mainly increased Γ ." The interposer layer can influence G by changing the crack pattern (i.e., changing Z in the equation $G=Zh_m\sigma^2/E_m$). However, this influence on G is unlikely as big as previously shown in Figure 1b. Inspired by the reviewer's question, we revised the G slope of case 3 to reduce its discrepancy with that of case 1 (Figure R1).

One sentence in the manuscript was revised as "For the case 3 and 4 in Fig. 1b, the interposer layer mainly increased Γ ". Figure R1 was used as the new Figure 1b.

Fig. R1 (new Fig. 1b). Schematic plot for the crack driving force – energy release rate (G) as a function of total strain. Γ_1 and Γ_2 refer to the interfacial toughness between the module and substrate with and without the adhesive interposer layer, respectively. Case 1 to 4 refer to the modules integrated onto the homogeneous substrate without the interposer layer (1), onto the module-specific substrate without the interposer layer (2), the homogeneous substrate with the interposer layer (3), the module-specific substrate with the interposer layer (4). The hard and soft regions in case 4 were separately plotted.

Reviewer #2 (Remarks to the Author):

Authors have addressed my questions and concerns satisfactory. I recommend to publish this work in Nature Communications.

A: We thank the reviewer for the supportive recommendation.